# DISTRIBUTED SPECULATIVE INFERENCE (DSI): SPECULATION PARALLELISM FOR PROVABLY FASTER LOSSLESS LANGUAGE MODEL INFERENCE

**Nadav Timor**[*]
Weizmann Institute of Science

**Jonathan Mamou**
Intel Labs

**Daniel Korat**
Intel Labs

**Moshe Berchansky**
Intel Labs

**Oren Pereg**
Intel Labs

**Moshe Wasserblat**
Intel Labs

**Tomer Galanti**
Texas A&M University

**Michal Gordon**
Weizmann Institute of Science

**David Harel**
Weizmann Institute of Science

## ABSTRACT

This paper introduces *distributed speculative inference (DSI)*, a novel inference algorithm that is provably faster than speculative inference (SI) (Leviathan et al., 2023; Chen et al., 2023; Miao et al., 2024; Sun et al., 2025; Timor et al., 2025) and standard autoregressive inference (non-SI). Like other SI algorithms, DSI operates on frozen language models (LMs), requiring no training or architectural modifications, and it preserves the target distribution. Prior studies on SI have demonstrated empirical speedups over non-SI—but rely on sufficiently fast and accurate drafters, which are often unavailable in practice. We identify a gap where SI can be slower than non-SI if drafters are too slow or inaccurate. We close this gap by proving that DSI is faster than both SI and non-SI—given any drafters. DSI is therefore not only faster than SI, but also unlocks the acceleration of LMs for which SI fails. DSI leverages *speculation parallelism (SP)*, a novel type of task parallelism, to orchestrate target and drafter instances that overlap in time, establishing a new foundational tradeoff between computational resources and latency. Our simulations show that DSI is 1.29-1.92x faster than SI in single-node setups for various off-the-shelf LMs and tasks. We open-source all our code. [1]

## 1 INTRODUCTION

Generative language models (LMs) have demonstrated unprecedented success across various tasks (OpenAI et al., 2023; Li et al., 2023a; Andreas, 2022; Bubeck et al., 2023). Reducing the inference latency of these models is a critical challenge for improving downstream applications and enabling further test-time scaling (OpenAI et al., 2024; Muennighoff et al., 2025). Faster inference can also drive broader adoption by real-time applications, which often prioritize low latency over other objectives. With the growing availability of hardware and decreasing costs, effectively utilizing more computing power for faster inference is becoming increasingly important.

A promising approach to reducing LM inference latency is *speculative inference (SI)*, built upon the principles of Burton (1985). SI employs faster *drafter* models to predict likely token continuations, which are then *verified* concurrently using modern hardware's data parallelism, known as *batching* (e.g., on GPUs) (Stern et al., 2018). SI has been widely adopted in practice thanks to novel lossless verification methods, demonstrating empirical speedups of up to 4x over standard autoregressive inference (Leviathan et al., 2023; Chen et al., 2023; Miao et al., 2024; Sun et al., 2025; Timor et al., 2025) and increasing throughput in multi-request settings (Sadhukhan et al., 2025). The core

---

[*]**Correspondence:** `nadav.timor@weizmann.ac.il`

[1]`https://github.com/keyboardAnt/distributed-speculative-inference`

advantage of SI is that it can generate more than one token per forward pass of a given *target* language model.

However, SI relies on a sequential draft-then-verify process, where each verification must be completed before drafting new tokens. As a result, SI is beneficial only if the drafters are sufficiently fast and accurate. If the drafters are too slow or inaccurate, SI fails to provide a speedup—or is even slower than standard autoregressive inference. This fundamental limitation of SI has not been addressed in prior work.

**Contributions.** To overcome the fundamental limitation of speculative inference (SI) as a sequential algorithm, we introduce *distributed speculative inference (DSI)*, a novel inference algorithm that parallelizes SI by leveraging *speculation parallelism (SP)*, a new type of task parallelism. Unlike SI, which blocks drafting until verification is complete, DSI overlaps verification with drafting, transforming SI into a non-blocking algorithm and effectively hiding verification latency.

Our key contributions are:

- Introducing speculation parallelism (SP): A novel type of task parallelism that eliminates the blocking nature of SI by enabling concurrent verification and drafting using multiple instances of the target and drafter models.
- Provable speedup over SI and non-SI: We prove that DSI is always at least as fast as non-SI and is strictly faster than both SI and non-SI in expectation.
- Broader applicability: DSI accelerates inference even with drafters for which SI fails, making it effective for a wider range of LMs.
- Scalability to available hardware: DSI can orchestrate an arbitrary number of GPUs ($\geq 2$) by adjusting its *lookahead* hyperparameter.
- Empirical validation: Our simulations show that DSI is 1.29-1.92x faster than SI across various models and tasks in realistic single-node, multi-GPU setups.

## 2 PRELIMINARIES

Below we describe speculative inference and how to measure latency. For rigorous definitions of autoregressive language models (LMs) and next-token prediction, we refer the reader to Appendix B.

**Speculative Inference (SI)** is an approach for accelerating the inference of a *target* LM $f_m$ (e.g., a member of the GPT series). Such methods use faster LMs $f_1, \ldots, f_{m-1}$ that approximate the target model $f_m$ in order to reduce the total inference time. For example, Leviathan et al. (2023); Chen et al. (2023) may reduce the amount of time it takes to generate $N > 1$ tokens from target model $f_2$ given a prompt $x_{\leq 0}$ by using batching as follows. The inference starts by drafting $k$ tokens $x'_i := f_1(x'_{\leq i-1}) := f_1(x_{\leq 0} \oplus x'_1 \oplus \cdots \oplus x'_{i-1})$ for $i \in [k]$ and $1 \leq k < N$ using a faster drafter model $f_1$. Then, the algorithm sends the prompts $\{x'_{\leq i}\}_{i=0}^k$ altogether as one input batch to the target model $f_2$. The idea is to take advantage of the data parallelism that modern GPUs offer to compute the logits corresponding to the prompts $\{x'_{\leq i}\}_{i=0}^k$ in parallel, hence faster than computing these $k+1$ individual logits sequentially. Given the logits, the algorithm generates $[1, k+1]$ tokens without additional forward passes. By repeating this process, the algorithm can generate $N > k+1$ tokens.

Straightforward algorithms of speculative inference are typically *lossless in expectation*, i.e., they generate tokens from the same distributions as the target would generate without speculation. Naive algorithms of speculation guarantee returning the same tokens as the target (Gante, 2023; Spector & Re, 2023; Timor et al., 2025). More sophisticated algorithms of speculation might generate different tokens, but their generated tokens follow the distribution of the target (Leviathan et al., 2023; Chen et al., 2023; Miao et al., 2024; Sun et al., 2025; Timor et al., 2025).

To implement distributed algorithms for speculative inference, we use multiple **processors** or **servers**, which are hardware components capable of **executing threads**. Processors can compute forward passes and sample tokens from the output probability vectors and we assume that threads can run in parallel. When using DSI we will run sequences of drafter models $f_{j_1}, f_{j_2}, \ldots, f_{j_k}$, where the first model takes $x_{\leq 0}$ and returns some token $x_1^{j_1}$, the second takes $x_{\leq 0} \oplus x_1^{j_1}$ as a prompt and returns $x_2^{j_1, j_2}$, and so on. As such, in order to denote that a given thread is computing the output of $f_{j_k}$ on

a sequence $x_{\leq k-1}^{j_1, \ldots, j_{k-1}} := x_{\leq 0} \oplus x_1^{j_1} \oplus \cdots \oplus x_{k-1}^{j_1, \ldots, j_{k-1}}$, we denote $C_J$, where $J = (j_1, \ldots, j_k)$. When a thread $C_J$ computes an LM, we denote the output probability vector by $C_J[\text{prob}]$. If $C_J$ samples a new token from $C_J[\text{prob}]$, we denote this token by $C_J[\text{new}]$. For example, thread $C_J$ implementing equation 3 will have

$$C_J[\text{prompt}] := x_{\leq i}, \ C_J[\text{prob}] := f\left(C_J[\text{prompt}]\right) \text{ and } C_J[\text{new}] \sim C_J[\text{prob}].$$

Once a thread $C_J$ finishes sampling a new token, the thread outputs the concatenation of $C_J[\text{prompt}]$ and $C_J[\text{new}]$. Following the example in equation 3, we have

$$C_J[\text{return}] := C_J[\text{prompt}] \oplus \left(C_J[\text{new}]\right) := (x_{\leq 0}, x_1, \ldots, x_{i+1}).$$

A new thread that was initiated by $C_J$ is denoted by $C_{J \oplus (j)}$, where $J \oplus (j)$ is the concatenation of $J$ and $(j)$. The set of all the threads that originate from $C_J$ is $\{C_{J \oplus J'} : J' \text{ is a nonempty tuple}\}$. We assume that terminating a concurrent thread terminates all the threads that originate from it.

**Time** in this paper is the *wall time*. We measure the time that passes from the initiation of a *task* until its termination. A task is a nonempty set of threads, denoted by $\{C_J : J \in \mathfrak{J}\}$. Its time is

$$T_{\text{wall}}\left[\{C_J\}_{J \in \mathfrak{J}}\right] := \max_{J \in \mathfrak{J}}\left(\text{Timepoint } C_J \text{ finishes}\right) - \min_{J \in \mathfrak{J}}\left(\text{Timepoint } C_J \text{ starts}\right).$$

When a task consists of a single thread, we omit the curly brackets, namely,

$$T_{\text{wall}}\left[C_J\right] := T_{\text{wall}}\left[\{C_J\}\right] \text{ where } |\{C_J\}| = 1.$$

Note that two threads, denoted by $C_J$ and $C_{J'}$, may run concurrently and overlap in time. Hence, it is possible that $\max\{T_{\text{wall}}\left[C_J\right], T_{\text{wall}}\left[C_{J'}\right]\} \leq T_{\text{wall}}\left[\{C_J, C_{J'}\}\right] < T_{\text{wall}}\left[C_J\right] + T_{\text{wall}}\left[C_{J'}\right]$. However, if $C_J$ and $C_{J'}$ do not overlap in time, then $T_{\text{wall}}\left[\{C_J, C_{J'}\}\right] \geq T_{\text{wall}}\left[C_J\right] + T_{\text{wall}}\left[C_{J'}\right]$.

## 3 DISTRIBUTED SPECULATIVE INFERENCE

This section presents a theoretically sound orchestration framework for parallelizing SI (Leviathan et al., 2023; Chen et al., 2023; Miao et al., 2024; Timor et al., 2025) that is essentially decoupled from the underlying computation of forward passes. Our method applies to any fixed number of processors ($\geq 2$), as shown later. Initially, we introduce a naive version of our approach, which assumes access to a sufficiently large number of processors, ensuring that threads never need to wait.

Before presenting our algorithm, we first discuss the limitations of existing SI methods. SI reduces a target forward whenever a draft token is accepted. With an accurate and fast drafter, SI can significantly cut the number of target forwards, potentially speeding up the inference compared to non-SI. For instance, if on average one draft token is accepted per iteration, the number of target forwards drops to half since the average target forward generates two tokens. As long as the total drafting latency is less than the saved latency from reduced target forwards, SI offers a speedup over non-SI. The primary limitation of existing SI methods lies in their sequential nature. Each SI iteration requires a target forward, and the next iteration only begins after the current one is completed. Therefore, SI with sufficiently slow or inaccurate drafters is slower than non-SI, even if reducing the number of target forwards. However, we observe that the verification of each SI iteration is not inherently sequential and could be parallelized.

Previous works on SI exemplified speedups using drafters that run within 1-5% of the time (compared to the target model), and iterations of 2-5 draft tokens (namely, `lookahead` $\in [2, 5]$). We can calculate the maximum speedup of our proposed method compared to SI by Amdahl's law, as follows. Assume the drafter is perfectly accurate. In that case, our method hides all the verification latency such that the overall end-to-end latency remains only the drafting latency. For drafters of 1-5% latency and `lookahead` $\in [2, 5]$, our proposed parallelization leads to a theoretical speedup of 4x-50x compared to SI.

Figure 1 and Table 1 illustrate potential speedups of our proposed method compared to SI and non-SI, given a drafter of 14% latency and `lookahead` $= 1$. Larger `lookahead` values (as often used in practice) yield even larger theoretical speedups.

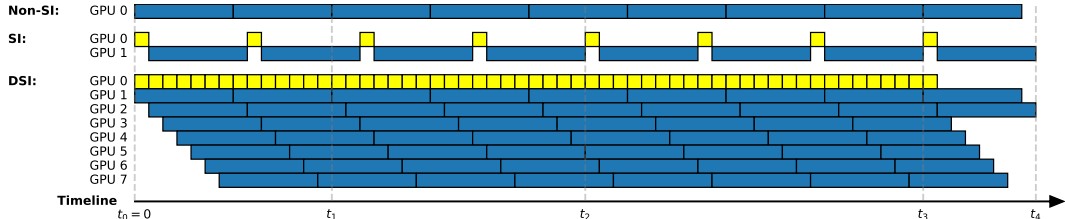

Figure 1: Illustration of the timeline for DSI, SI, and autoregressive inference (non-SI). Blue and yellow mark the forward latency of the target and drafter, respectively. In this example we have `lookahead = 1`, namely, the drafter generates a single token in every yellow square. Non-SI and SI are both sequential: each of their iterations ends with a target forward, and this target forward must be completed before the next iteration can start. In DSI, target forwards are not necessarily blocking as in SI and non-SI. While DSI works for any given number of GPUs ($\geq 2$), here it orchestrates eight GPUs.

Table 1: The number of tokens that non-SI, SI, and DSI generate according to Figure 1. In the worst case, all the draft tokens (yellow in Figure 1) are rejected. In the best case, all the drafts are accepted. The number of tokens generated by DSI is greater than or equal to the number of tokens generated by SI and non-SI in all cases, at any time, in expectation.

|  |  | $t_1$ | $t_2$ | $t_3$ | $t_4$ |
|---|---|---|---|---|---|
| | non-SI | **2** | **4** | **8** | **9** |
| Worst case | SI | 1 | **4** | 7 | 8 |
| | DSI | **2** | **4** | **8** | **9** |
| | non-SI | 2 | 4 | 8 | 9 |
| Best case | SI | 2 | 8 | 14 | 16 |
| | DSI | **8** | **26** | **50** | **58** |

### 3.1 METHOD OVERVIEW

Consider the task of computing $N$ output tokens autoregressively from a target model $f_m$ given a prompt $x_{\leq 0}$. We have a set of faster drafter models, $f_1, \ldots, f_{m-1}$, that are all faster than $f_m$ (as defined in Assumption 2). Our goal is to compute $x_i = f_m(x_{\leq i-1})$ for all $i \in [N]$. Appendix C provides a detailed, step-by-step explanation of Algorithm 1.

**Acceptance rate.** Lines 8 and 10 of Algorithm 1 terminate any thread (and its descendants) that returns a token that does not match the token returned by the current verifier. We say that this draft token is *rejected*. Given a target, a drafter, and an input prompt, we define the *acceptance rate* to be the probability of accepting the draft token. To increase the acceptance rate, we can replace the strict exact-matching (lines 8, 10) with relaxed methods for rejecting drafts. For example, applying the procedures proposed in Leviathan et al. (2023); Chen et al. (2023); Miao et al. (2024); Timor et al. (2025) increases the acceptance rate while maintaining the distribution of the outputs of the target model (namely, lossless in expectation).

**Speculation parallelism (SP).** In essence, DSI offers a new type of task parallelism we name *speculation parallelism (SP)* that orchestrates instances of the target and drafters to overlap in time. *Speculation parallelism degree (SP degree)* is the maximal number of target servers (namely, servers dedicated to computing the target model) used at the same time. DSI parallelizes all of the non-sequential parts of SI. Unlike SI, where verifying the drafts of the current iteration is sequential such that the verification blocks the algorithm from drafting the next iteration, DSI runs verifications on additional threads to hide their latency. In DSI, verifications contribute to the overall latency only when they reject a token. Rejecting a token in DSI triggers a synchronization mechanism terminating threads that received a rejected token (line 8), ensuring the output tokens are all accepted. The portion of the inference that DSI parallelizes tends to the expected acceptance rate as the number of generated tokens grows to infinity. For example, in the simple case where we have a single drafter with 80% acceptance rate, DSI effectively parallelizes 80% of the work such that the expected speedup is 5x by Amdahl's law for generating $N \to \infty$ tokens.

---

**Algorithm 1** Distributed Speculative Inference (DSI) of $N$ tokens

---

**Require:** A prompt $x_{\leq 0}$, and $m$ autoregressive models, $f_1, f_2, \ldots, f_m$.

1: $v = 1$.
2: **initiate** $m$ threads $C_{(1)}, \ldots, C_{(m)}$ such that $C_{(j_1)}$ generates the token $x^{j_1} \sim f_{j_1}(x_{\leq 0})$ for all $j_1 \in [m]$ concurrently.
3: **label** thread $C_{(m)}$ as the current verifier.
4: **ONCE** any thread $C_{J \oplus (j)}$ finishes to generate a token, namely, sampled $C_{J \oplus (j)}[\text{new}] \sim f_j\left(C_{J \oplus (j)}[\text{prompt}]\right)$:
5:  **if** $|J| + 1 < N$ **then**
6:    **initiate** $m$ threads, $C_{J \oplus (j,1)}, C_{J \oplus (j,2)}, \ldots, C_{J \oplus (j,m)}$, to generate a token concurrently and respectively from $f_1, f_2, \ldots, f_m$.
7:    **if** $C_{J \oplus (j)}$ is the current verifier thread **then**
8:      **terminate** all threads $C_{J \oplus (j')}$ (and their descendant threads) that sampled a different token than $C_{J \oplus (j)}$.
9:      let $j^* = \arg \min_{j' \in [m]} \{j' \mid C_{J \oplus (j')}[\text{new}] = C_{J \oplus (j)}[\text{new}]\}$.
10:      **terminate** all threads $C_{J \oplus (j')}$ (and their descendant threads), where $j' > j^*$.
11:      **label** $C_{J \oplus (j^*, m)}$ as the current verifier.
12:      **update** $v = v + 1$.
13:      **if** $C_{J \oplus (j^*, m)}$ has already finished **then**
14:        go back to step 7 with $J = J \oplus (j^*, m)$.
15:      **end if**
16:    **end if**
17:  **else if** the last entry of $J \oplus (j)$ equals $m$ (i.e., $j = m$) **then**
18:    **return** $C_{J \oplus (j)}[\text{return}]$.
19:  **end if**
20: **end ONCE**

---

**Lookahead.** While the abstract version of DSI described in Algorithm 1 takes advantage of a sufficiently large number of servers, in practice we typically have a fixed number of servers. We can deploy DSI on an arbitrary number of servers ($\geq 2$) by selecting a sufficiently large `lookahead` hyperparameter, as elaborated in Appendix D. The `lookahead` is defined as the number of draft tokens in every verification task sent to a target server. The `lookahead` in Algorithm 1 is set to 1 for simplicity, but can be arbitrarily large. For example, verifying every five draft tokens (`lookahead` = 5) instead of one (`lookahead` = 1) is a standard configuration of SI methods in practice. In DSI, larger `lookahead` values decrease the frequency at which verification tasks are sent to the target servers, hence decreasing the required SP degree. Given an SP degree, the `lookahead` must be sufficiently large to satisfy the following inequality, ensuring that verification tasks never wait to be processed by a target server.

$$\left\lceil \frac{(\text{target latency})}{(\text{lookahead}) \cdot (\text{drafter latency})} \right\rceil \leq \text{SP} \tag{1}$$

Smaller SP degrees or faster drafters require selecting larger `lookahead` values. For example, given a single drafter of 5% latency and $\text{SP} = 4$, having `lookahead` = 5 is sufficient. Assuming a single drafter that runs on a single processing unit, the maximum number of required processing units is $1 + \left\lceil \frac{1}{5 \cdot 0.05} \right\rceil = 5$. If more than five processing units are available, we can select a smaller `lookahead` value, yielding verification tasks more frequently. In general, selecting the minimum `lookahead` value that satisfies Equation 1 is the optimal choice, allowing DSI to detect rejections (line 8) earlier. Using an SP degree such that $\text{SP} = \left\lceil \frac{\text{target latency}}{\text{drafter latency}} \right\rceil$ reaches the maximum expected speedup, and any larger SP degree cannot speed up the inference because there will be more target servers than verification tasks that can be processed in parallel.

**Resource contention.** In practice, resource contention might occur when multiple threads compete for the same hardware resources, such as memory bandwidth, data transfer channels, or CPU cores used for orchestration. By selecting the minimal `lookahead` that ensures the required SP degree is supported by the available resources (namely, satisfying Equation 1), we can guarantee that the algorithm runs efficiently in practice, without significant resource contention, because the verification requests are sent to the target server at different times, and the responses are expected in staggered timings.

**Model parallelism (MP).** DSI introduces a new way to parallelize that differs from tensor parallelism (TP) and pipeline parallelism (PP). DSI can employ TP, PP, or a combination of both TP and PP. To simplify, we say that MP is any such parallelism combination, including TP, PP, or TP+PP. MP speeds up the computation of forwards, and SI reduces the number of target forwards. Their combination (SI+MP) reduces the number of target forwards and accelerates each target forward. DSI further reduces the number of target forwards *contributing* to latency because DSI hides the latency of verification tasks by computing them in parallel. Below we compare SP and MP, providing a simple example demonstrating that SP outperforms MP given the same computing budget.

In DSI, target forwards contribute latency only if they reject a draft. Below is a simple example of comparing MP and SP. Given a drafter of 10% latency, we can set `lookahead = 2` to allow DSI to run over a single node of only 6 GPUs (5 for the target and 1 for the drafter). Let $a$ be the acceptance rate of the drafter. The expected number of target forwards that DSI eliminates by hiding is $a^2$. For example, for a drafter with an 80% acceptance rate, only 36% of the target forwards contribute to latency (in general, $1 - a^{\text{lookahead}}$). Under the same computing budget (MP=5), MP must accelerate the target forwards by 2.78x or more to become faster than DSI. However, MP is ineffective for certain hardware setups, model architectures and sizes, while DSI remains effective.

DSI could be naturally combined with MP to accelerate the underlying forwards, requiring no changes to the algorithm, because DSI offers an orchestration algorithm agnostic to the underlying computation of forwards. Since DSI can orchestrate multiple nodes, it unlocks setups with sufficiently large SP degrees so that there is no theoretical tradeoff between DSI and MP. Combining DSI with MP methods can possibly further accelerate the inference in both single- and multi-node setups. All the foundational concepts of such an implementation of DSI are covered in this paper.

**KV cache.** We can view DSI as an orchestration algorithm that constructs and verifies token trees on the fly. DSI is decoupled from the underlying computation of forwards, including KV cache management, both in theory and in practice. Each server maintains its own KV cache. The servers collaboratively process a token tree with shared prefixes. Synchronizations occur at every draft rejection.

Efficient KV cache management of token trees has already been developed in SpecInfer, where tree paths can share common prefixes (Miao et al., 2024). Practitioners can apply SpecInfer's KV cache management as-is to achieve the expected speedups reported in this paper. While it might require some engineering effort to implement SpecInfer's KV cache management, it is a solved research problem and has been shown to add negligible latency.

## 3.2 THEORETICAL RESULTS

Next, we formally state that DSI (Algorithm 1) is strictly lossless, always returning the correct sequence of tokens $x_1, \ldots, x_N$, runs at least as fast as non-SI, and runs faster than both non-SI and SI in expectation. Before presenting our main theoretical results, we outline the assumptions used in the analysis. The proofs are provided in Appendix E.

**Assumption 1.** *We assume the existence of a (universal) constant $c > 0$ such that, for any input prompt $x_{\leq 0}$ and model index $j \in [m]$, we have:*

$$T_{wall}\left[\text{computing } f_j\left(x_{\leq 0}\right)\right] \in (0, c) \quad and \quad T_{wall}\left[\text{sampling } x \sim f_j\left(x_{\leq 0}\right)\right] = 0.$$

**Assumption 2.** *We assume that for all $j \in [m-1]$, $f_j$ is faster than $f_m$ (denoted $f_j \preceq f_m$) in the following sense* $\max_{x_{\leq 0}} T_{wall}\left[\text{computing } f_j\left(x_{\leq 0}\right)\right] \leq \min_{x_{\leq 0}} T_{wall}\left[\text{computing } f_m\left(x_{\leq 0}\right)\right]$.

**Assumption 3.** *We assume that $T_{wall}\left[\{C_{(j_1, \ldots, j_i)}\}_{i=1}^k\right] = \sum_{i=1}^k T_{wall}\left[C_{(j_1, \ldots, j_i)}\right]$.*

The first assumption asserts that computing the output of any model takes a non-zero, bounded amount of time, and sampling a token from the output probabilities takes a negligible amount of time. The second assumption asserts that each drafter model runs faster than the target model, for any given input prompt. The third assumption asserts that computing $x_k^{j_1, \ldots, j_k}$ takes the time of first computing $x_1^{j_1}$, then $x_2^{j_1, j_2}$, and so forth, up to $x_k^{j_1, \ldots, j_k}$, with no delays.

The following theorem suggests that our method returns tokens from the same distributions as those the target would generate without speculation, and is at least as fast as iteratively applying the target model itself.

**Theorem 1.** *Under Assumptions 1, 2 and 3, Algorithm 1 returns the same output and runs at least as fast as running the target model itself without speculative inference (SI).*

**Theorem 2.** *Under Assumptions 1, 2 and 3, Algorithm 1 runs at least as fast as SI in expectation.*

The advantage of Algorithm 1 lies in its concurrency. The following proposition shows how DSI can accelerate the inference of a given target model using a drafter model that is faster than the target model and returns the correct output with high probability.

**Proposition 1.** *Suppose we have a drafter model $f_1$, a target model $f_2$ and a prompt $x_{\leq 0}$. Assume that $f_1$ requires $t_1$ time units to compute each of its outputs, and $f_2$ requires $t_2$ time units, where $t_2 > t_1$. Assume that given the prompt $x_{\leq i} = x_{\leq 0} \oplus x_1 \oplus \cdots \oplus x_i$, the probability that $f_1$ returns the (correct) token $x_{i+1}$ is $p$. Then, the expected time it takes Algorithm 1 to calculate the correct output is at most $t_1 p(N-1) + t_2((1-p)(N-1) + 1)$ time units, compared to the $t_2 N$ time units required if we were to compute $f_2$ without speculative inference.*

## 4    EMPIRICAL RESULTS

DSI can orchestrate any fixed number of processors ($\geq 2$) by selecting a sufficiently large `lookahead` value. While the theoretical guarantees in section 3.2 hold for both single- and multi-node setups, our experiments are confined to single-node scenarios with at most eight GPUs.

Configuring DSI for any given number of GPUs ($\geq 2$), target model, and drafter, requires calculating the SP degree by selecting the allocation of the available GPUs, then selecting the `lookahead` to be the minimal number satisfying Equation 1. In its simplest nontrivial setup, DSI orchestrates a single node with two GPUs, implementing a target server on one GPU and a drafter server on the other. More advanced setups involve more GPUs, potentially on other nodes, or combine DSI with other parallelism techniques so that servers can utilize multiple underlying GPUs. For example, a node with eight GPUs can run eight servers, each using one GPU, or four servers, each using two GPUs, employing model parallelism (MP) with tensor parallelism (TP) or pipeline parallelism (TP). To maximize the expected speedup, we select the minimal `lookahead` value that satisfies Equation 1 so that verification tasks never wait to be processed by a target server. Some models require more than one GPU to avoid memory offloading to slower memories (like the host's CPU memory or hard disk). For example, given seven GPUs and a target model that requires two of the given GPUs to run without offloading (namely, MP $\geq 2$), the SP degree is at most three, assuming that the drafter can run on a single GPU. With an SP degree of three, we will select the `lookahead` to be the minimal number satisfying Equation 1. If the drafter forwards take 5% of the target forwards, the ratio between their latencies is 20, hence, the minimum `lookahead` value guaranteeing no waiting is 7.

Since our experiments focus on single-node setups, we implemented DSI as a multithreading system. We implemented DSI in Python, using Python's built-in thread pool to manage operating system (OS) threads. These OS-level threads share CPU resources similarly to other real-world multi-threaded systems. This ensures that real-world thread management latencies, such as context switching, thread creation, and scheduling delays, are fully incurred in the experiments. For multi-node setups, implementing DSI as a multiprocessing system with a Message Passing Interface (MPI) would be more appropriate than multithreading.

In both single- and multi-node setups, DSI can employ a thread pool design pattern, where verification tasks are sent to a pool of servers computing the target model. The size of this target pool is, by definition, the SP degree. Our implementation is based on a thread pool of targets and a single drafter server. In all our experiments, DSI has an SP degree of less than or equal to seven and one drafter server, where each server employs a single GPU.

Our implementation of DSI and the code for running the experiments are agnostic to the underlying hardware, and have been open-sourced, employing high software development standards with thousands of tests, ensuring complete reproducibility of the reported results and enabling further research.

Measuring the actual speedups of DSI for a node with eight GPUs requires access to such a node. Due to budget constraints, instead of a node with eight GPUs, we only had access to one GPU. To evaluate DSI over a node with eight GPUs without access to such hardware, we adjusted the DSI

implementation accordingly. While DSI incurs all the real-world latencies typical of multithreading systems (e.g., latencies of managing OS threads), each call to compute the forward pass of an LM was replaced by a `wait` command. The `wait` command blocks the thread for a duration that matches the actual latency.

To ensure realistic `wait` times, we conducted a separate experiment to estimate the Time to First Token (TTFT) and Time Per Output Token (TPOT) for each model and dataset. These TTFT and TPOT values were then used to set the `wait` times in the main experiment. To estimate the acceptance rate for each combination of ⟨target, drafter, dataset⟩, we performed another separate experiment and plugged in the approximated acceptance rate in the main experiment. Appendices F.1 and F.2 provide detailed explanations of the independent experiments to estimate the TTFT, TPOT, and acceptance rate.

The results of our main experiment, detailed in Table 2, affirm Theorems 1 and 2, demonstrating that DSI outperforms SI (Leviathan et al., 2023; Chen et al., 2023) in practical settings across various models and well-known datasets. Overall, DSI consistently outperforms SI across all models and tasks. Table 2 also specifies the latency and acceptance rate estimates used in each configuration.

Table 2: DSI speedups over SI for various off-the-shelf target/drafter pairs. We observe that DSI outperforms SI consistently across all models and tasks.

| Target | Drafter | Dataset | Target Latency (ms) | Drafter Latency (ms) | Drafter Latency (%) | Acceptance Rate (%) | Speedup DSI vs. SI |
|--------|---------|---------|---------------------|----------------------|---------------------|---------------------|---------------------|
| Starcoder-15B | Starcoder-168M | HumanEval | 20.6 | 6.8 | 32.3 | 93 | 1.92x |
| Starcoder-15B | Starcoder-168M | MBPP | 21.0 | 6.8 | 32.9 | 90 | 1.66x |
| Phi3-14B | Phi3-4B | Alpaca | 49.6 | 33.4 | 67.4 | 87 | 1.60x |
| Phi3-14B | Phi3-4B | HumanEval | 52.1 | 34.0 | 65.3 | 95 | 1.41x |
| Phi3-14B | Phi3-4B | CNN-DM | 52.4 | 34.6 | 66.0 | 93 | 1.39x |
| Phi3-14B | Phi3-4B | MBPP | 52.2 | 34.3 | 65.8 | 94 | 1.37x |
| Vicuna-13B | Vicuna-68M | CNN-DM | 37.7 | 2.5 | 6.5 | 63 | 1.47x |
| Vicuna-13B | Vicuna-68M | Alpaca | 33.3 | 2.5 | 7.4 | 58 | 1.41x |
| Vicuna-7B | Vicuna-68M | CNN-DM | 29.4 | 2.5 | 8.4 | 67 | 1.29x |
| Vicuna-7B | Vicuna-68M | Alpaca | 26.0 | 2.5 | 9.5 | 59 | 1.70x |

The reported speedup of DSI relative to SI ("Speedup DSI vs. SI" in Table 2) is the ratio between their estimated end-to-end latencies (wall time), including prefilling and decoding latency, but excluding tokenization latency. The end-to-end latency is estimated as follows: we generate 50 tokens using each target-drafter pair on each dataset, employing real-world forward latencies and acceptance rate values from independent experiments (elaborated below). Each combination of ⟨target, drafter, dataset⟩ is run on multiple `lookahead` values (specifically, 1, 5, and 10, because this range has been shown in previous works on SI to be effective). For the DSI run, we further restrict the `lookahead` values to ensure we only consider configurations for which DSI could have been deployed on a single node with up to eight GPUs, assuming the drafter runs on a single GPU. That is, we only use `lookahead` $\in \{1, 5, 10\}$ if this `lookahead` value satisfies Equation 1 for SP $= 7$.

In all our experiments, the models and datasets were downloaded from the Hugging Face Hub and used as-is. We used four well-established datasets to estimate forward latencies and acceptance rates, spanning various tasks: text summarization using CNN Daily Mail (Hermann et al., 2015); instruction-following using Alpaca (Taori et al., 2023); code generation using MBPP and HumanEval (Austin et al., 2021; Chen et al., 2021). Appendices F.5 and F.6 provide a complete description of the models, datasets, and examples of prompts.

## 4.1 ABLATION VIA OFFLINE SIMULATION

The main experiment above (Table 2) can be viewed as an "online" experiment because it employs thread pools and measures the overall wall time including real-world latencies of multithreading systems. This section presents a complementary "offline" experiment that simulates the inference algorithms by simply summing the forward latencies without thread pools, assuming zero multithreading latencies. The offline experiment is important for two reasons: (i) it decouples the implementation details of DSI from the theoretical analysis, and (ii) it allows us to explore a much larger space of configurations within a constrained computational budget.

Figure 2 presents the results of this offline simulation, measuring the pairwise speedups (or slow-downs): SI compared to non-SI, DSI compared to SI, and DSI compared to non-SI. Since SI is slower than non-SI in some configurations, we have included Figure 2(d) as an additional comparison that shows DSI speedups relative to the faster of the two algorithms—SI or non-SI—for any given configuration. Figure 2(d) helps identify configurations where DSI achieves the highest speedup. It demonstrates that, unlike SI, our method introduces no slowdown compared to non-SI and consistently accelerates inference.

As shown in Figure 2(a), to achieve a speedup with SI compared to non-SI, the acceptance rate of the drafter must at least match the latency of the drafter model, which corresponds to the non-pink region in the figure. This means that the SI algorithm cannot speed up the inference if the acceptance rate of the drafter is not sufficiently high for a given latency, corresponding to the pink region in the figure. Conversely, in Figure 2(b), we observe that DSI consistently speeds up inference time, regardless of the latency and acceptance rates of the drafter. This provides our method with much greater flexibility and robustness when selecting drafters for a given target model. In Figure 2(c), we observe that DSI is faster than non-SI for all configurations for which non-SI is faster than SI. Finally, to obtain a comprehensive view of the inference speedup achieved by DSI, in Figure 2(d), we compare the performance of DSI with the minimum runtime for any configuration between SI and non-SI.

The heatmaps represent millions of data points, where each point corresponds to a different configuration. Since offline simulations are insensitive to the real-world latencies of multithreading systems (e.g., context switching), an offline simulation for a particular configuration could be run in parallel with other offline simulations. This approach of parallelizing the experiments—rather than running them sequentially—makes it feasible to scale up the number of configurations explored within a constrained computational budget and provide comprehensive heatmaps of the expected speedups. In contrast, for online experiments, each of the millions of configurations represents a distinct online run that necessitates a separate environment.

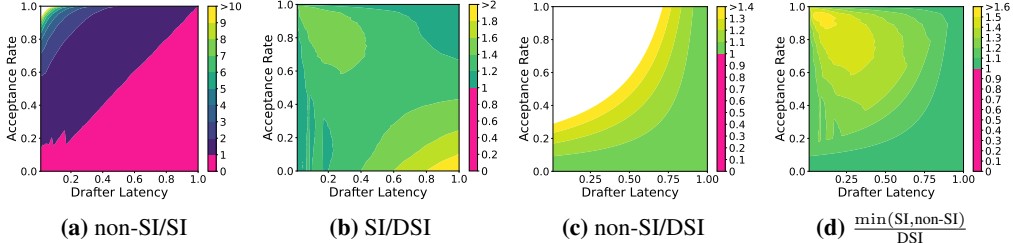

**(a)** non-SI/SI    **(b)** SI/DSI    **(c)** non-SI/DSI    **(d)** $\frac{\min(\text{SI},\text{non-SI})}{\text{DSI}}$

Figure 2: Expected pairwise speedups (or slowdowns) of DSI, SI, and non-SI. Each heatmap is labeled "X/Y" and plots the ratio between the run time of algorithm X and the run time of algorithm Y. The run time of each algorithm is computed by summing the latencies of all the forward passes and intentionally ignoring additional real-world latencies of multithreading systems like context switching, allowing us to decouple the implementation details from the theoretical analysis. **(a)**: SI is slower than non-speculative inference (non-SI) when the drafter is either slow or inaccurate enough (pink marks slowdowns). **(b, c, d)**: DSI is faster than speculative inference (SI) and non-speculative inference (non-SI) for all configurations of non-zero acceptance rate. DSI is never slower than either SI or non-SI for all configurations. **(d)**: DSI is up to 1.6x faster than the baseline algorithm, where the baseline is the faster between SI and non-SI for each configuration.

Appendix F provides additional implementation details about the experiments. We open-source the code for all the simulations.

## 5 RELATED WORK

Research on SI has expanded the framework, including dynamically controlling the number of draft tokens per iteration—a technique widely adopted in practice (Mamou et al., 2024; Liu et al., 2024a; Gante, 2023)—and exploring various other directions (Li et al., 2024; Cai et al., 2024; Sun et al., 2024b; Liu et al., 2024b; Zhou et al., 2024; Zafrir et al., 2024; Narasimhan et al., 2025; Bachmann et al., 2025). However, in prior works, computing target forward passes remains a blocking operation,

limiting the algorithm from processing tokens in later positions and leaving the fundamental limitation of SI as a sequential algorithm unaddressed.

PEARL is a recent[2] extension to SI, demonstrating that drafting in parallel to verifying speeds up non-SI and vanilla SI by up to 4.43x and 1.5x, respectively (Liu et al., 2025). Their empirical results highlight the advantage of DSI in breaking the draft-then-verify sequential nature of SI. However, PEARL suffers from a fundamental limitation: it remains a sequential algorithm because it can only process tokens of the next SI iteration, unlike DSI, which can process tokens of any future iteration. Their algorithm employs a heuristic (controlling whether to verify the first draft token of every iteration) and has no theoretical guarantees to speed up SI or non-SI. In fact, PEARL is slower than non-SI if the drafters are too slow or inaccurate, unlike DSI. Since PEARL cannot orchestrate more than one instance of the target model and one instance of the drafter, it offers limited scalability to hardware setups, unlike DSI, which can orchestrate an arbitrary number of GPUs ($\geq 2$). As a result, PEARL underutilizes the available hardware unless it runs with lookahead $= \frac{\text{target latency}}{\text{drafter latency}} > 1$, and is therefore strictly slower than DSI with a smaller lookahead, in expectation.

For other related work beyond SI, see Appendix A.

## 6 DISCUSSION

This work proposes a method to reduce the run time of speculative inference algorithms by taking advantage of an arbitrary number of additional multiple processing units (e.g., GPUs). We have shown that despite the wide adoption of SI algorithms, they can end up slowing the inference of language models in various practical settings, when the drafters are insufficiently accurate or fast. We showed that by taking advantage of at least one additional GPU, we can design a speculatively inference algorithm that provably reduces the inference time of both non-SI and SI algorithms. Our simulations affirm our theory, indicating significant speedups over both SI and non-SI for all possible configurations given a single node with up to eight GPUs. In essence, this work paves the way to additional SI algorithms that can orchestrate multiple processing units at the same time via *speculation parallelism* (SP).

We introduce *distributed speculative inference* (DSI) and show it is faster than SI and non-SI for all possible configurations by theoretical analysis and experiments. While the theoretical guarantees hold for both single- and multi-node setups, our experiments focus on single-node scenarios with up to eight GPUs with an SP degree $\leq 7$. Due to budget constraints, we adjusted our implementation of DSI to simulate an access to such a node rather than running on a physical node with eight GPUs. Nevertheless, the empirical results are realistic because DSI is implemented as a multithreading system, incurring all real-world latencies of such systems (e.g., context switching), and all the real-world GPU-related latencies are based on independent experiments accessing a GPU.

**Future Work.** DSI's key strengths lie in its use of *speculation parallelism* to reduce latency through parallelized verification, and its ability to achieve lossless speedups without modifying model architectures. Building on these, future work should focus on evaluating DSI with LMs that require multiple GPUs to avoid memory offloading (namely, MP degree $> 1$). While most state-of-the-art models can run on a single GPU (via compression, in a lower precision, etc.), we do see a trend towards even larger LMs. Since larger LMs are often slower, DSI's parallelism could be particularly effective. Although multi-node inference is not yet a common setup, testing DSI in realistic multi-node environments, could unlock its potential despite communication latencies.

### ACKNOWLEDGMENTS

We thank Intel Labs for funding this research.

This work was partially funded by the Israel Science Foundation (ISF grant 3698/21). Additional support was provided by a research grant to David Harel from Louis J. Lavigne and Nancy Rothman, the Carter Chapman Shreve Family Foundation, Dr. and Mrs. Donald Rivin, and the Estate of Smigel Trust. Michal Gordon-Kiwkowitz's contribution was conducted as part of her consulting position at the Weizmann Institute of Science. She is also affiliated with the School of Computer Science at

---

[2]Their work was released on arXiv and GitHub after ours and is published concurrently at ICLR.

Holon Institute of Technology and has recently moved to Indiana University Bloomington. Part of Tomer Galanti's contribution was conducted while he was at MIT.

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

# A    OTHER RELATED WORK

Beyond SI, other recent efforts to reduce the inference latency of LMs can be classified into two main categories by their approach to hardware utilization. The first category focuses on accelerating the inference by using more computing power. It includes data parallelism (DP) and model parallelism (MP) methods of different types, such as pipeline parallelism (PP), tensor parallelism (TP) (Narayanan et al., 2021), context parallelism (Li et al., 2023b; Yang et al., 2024), and expert parallelism (Rajbhandari et al., 2022). Such partitioning over multiple processors can speed up memory-bounded inference setups, for example, by avoiding memory offloading. They are often effective in increasing the inference throughput and supporting larger context lengths. However, the typical LM architectures necessitate autoregressive inference. Such inference is inherently sequential with only limited (if any) MP opportunities, depending on the LM architecture. Since the portion of the inference that could be parallelized is small (if any) in typical LMs, the potential speedup of MP methods is limited, by Amdahl's law (Amdahl, 1967; Rodgers, 1985). Hence, while DP methods can increase the inference throughput, they remain inherently sequential. Furthermore, as the parallelisms above shard over more processors, the communication overhead increases, which can lead to diminishing returns, limiting the number of processors they can effectively utilize.

The second category focuses on making LMs use less computing resources or better utilize the same resources. This includes post-training compression through pruning (e.g., (Frantar & Alistarh, 2023; Sun et al., 2024a; Ma et al., 2023)), knowledge distillation (e.g., Hinton et al. (2015); Gu et al. (2024)), quantization (e.g., (Hubara et al., 2018; Frantar et al., 2023; Lin et al., 2024; Yao et al., 2024; Dettmers et al., 2022)), low-rank factorization (e.g., (Hsu et al., 2022; Xu et al., 2023)), early exiting (e.g., (Schuster et al., 2022; Kim et al., 2023; Elbayad et al., 2020; Bapna et al., 2020; Schuster et al., 2021)), and alternative architectures (e.g., (Cai et al., 2024; Li et al., 2024; Zhang et al., 2024b;a; Xiao et al., 2024; Gu & Dao, 2024)). While these solutions are useful in practice, they often require modifications to the model architecture, changes to the training procedure and re-training of the models, without guaranteeing identical outputs. Despite reducing the inference time, these methods often have a significant drawback of degrading the output quality. There are also solutions that preserve the output quality, such as kernel optimizations (Dao et al., 2022; Dao, 2024), but they highly depend on the hardware and therefore are not always available or even feasible.

# B    EXTENDED PRELIMINARIES

**Autoregressive language models (LMs)** are deterministic, real-valued multivariate functions. An input to an LM is a sequence of vectors of dimension $n_{\text{vocab}}$. We call these vectors *tokens*, and the sequence a *prompt*. LMs output a real-valued vector of dimension $n_{\text{vocab}}$, also known as the *logits*. Since prompts may vary in length, we simplify the notation of the *forward pass* as follows: $f : \mathbb{R}^{* \times n_{\text{vocab}}} \to \mathbb{R}^{n_{\text{vocab}}}$.

**Self-Attention LMs** are LMs with a pre-defined context length $n_{\text{ctx}}$ (Vaswani et al., 2017). Hence, we represent the forward pass of such LMs in the following manner: $f : \mathbb{R}^{n_{\text{ctx}} \times n_{\text{vocab}}} \to \mathbb{R}^{n_{\text{vocab}}}$. For example, GPT-2 and GPT-3 are Transformers with $n_{\text{vocab}} = 50257$, and context lengths $n_{\text{ctx}} = 1024$ and $n_{\text{ctx}} = 2048$, respectively (Radford et al., 2019; Brown et al., 2020). In this paper, all LMs are Self-Attention ones with pre-trained (frozen) parameters.

We extend the prompt notation such that prompts can have length $l \leq n_{\text{ctx}}$. Self-Attention LMs handle prompts of length $l < n_{\text{ctx}}$ by starting the input sequence with a prefix of $n_{\text{ctx}} - l$ tokens, followed by the $l$ given tokens. LMs ignore the prefix, either by zeroing (masking) the Attention parts corresponding to the prefix or by left-padding with dedicated tokens. In this paper, prompts of length $l < n_{\text{ctx}}$ are the non-masked, non-padded suffix of the input sequence of length $n_{\text{ctx}}$.

**Generating the next token** is the primary application of autoregressive LMs. This process consists of two steps: computing the forward pass of the LM and then selecting the next token based on the output. The selection can be deterministic or non-deterministic. Non-deterministic selection procedures apply the softmax function after the forward pass of LMs and sample from the resulting probability vector:

$$\text{softmax} : \mathbb{R}^{n_{\text{vocab}}} \to [0, 1]^{n_{\text{vocab}}} \text{ such that the entries sum to 1.} \tag{2}$$

For convenience, we denote the output probability vector by $f(x_{\leq i})$:

$$x_{i+1} \sim f(x_{\leq i}) := \text{softmax}(f(x_{\leq i})) := \text{softmax}(f(x_{\leq 0} \oplus x_1 \oplus \cdots \oplus x_i)), \qquad (3)$$

where $a \oplus b = (a, b)$ is the concatenation of the vectors $a$ and $b$ and $x_{\leq i} := x_{\leq 0} \oplus x_1 \oplus \cdots \oplus x_i$. For deterministic selection procedures, composing monotonic functions, such as softmax, is usually unnecessary. For example, the most likely next token is the $\arg\max$ of both the logits and the output of the softmax. Still, for convenience, we assume that LMs always output probability vectors. The sampling process in equation 3 is either deterministic (i.e., $x_{i+1}$ is a token with maximal probability) or random (achieved by randomly selecting $x_{i+1}$ from the distribution softmax $(f(x_{\leq i}))$).

## C  STEP-BY-STEP METHOD OVERVIEW

Consider the task of computing $N$ output tokens autoregressively from a target model $f_m$ given a prompt $x_{\leq 0}$. We have a set of faster drafter models, $f_1, \ldots, f_{m-1}$, that are all faster than $f_m$ (as defined in Assumption 2). Our goal is to compute $x_i = f_m(x_{\leq i-1})$ for all $i \in [N]$. To achieve this, we initiate $m$ threads, $C_{(1)}, \ldots, C_{(m)}$ (line 2 in Algorithm 1). Each thread, denoted as $(j_1)$, is responsible for computing $x_1^{j_1} = f_{j_1}(x_{\leq 0})$. Once a thread, $C_{(j_1)}$, finishes computation, we instantiate $m$ new threads, $C_{(j_1, j_2)}$, to calculate $x_2^{j_1, j_2} = f_{j_2}(x_{\leq 0} \oplus x_1^{j_1})$ for all $j_2 \in [m]$. In general, once we compute $x_{r-1}^{j_1, \ldots, j_{r-1}}$, we initiate $m$ new threads, $C_{(j_1, \ldots, j_{r-1}, 1)}, \ldots, C_{(j_1, \ldots, j_{r-1}, m)}$, to compute $x_r^{j_1, \ldots, j_r} = f_{j_r}(x_{\leq 0} \oplus x_1^{j_1} \oplus \cdots \oplus x_{r-1}^{j_1, \ldots, j_{r-1}})$ for all $j_r \in [m]$. This is captured in lines 4 and 6.

Once $C_{(m)}$ completes its computation and provides the correct value of the first output token $x_1^m = x_1$, we can verify which other threads, $C_{(j_1)}$, have accurately computed $x_1$. Any thread $C_{(j_1)}$ where $x_1^{j_1} \neq x_1$ is immediately terminated along with its descendant processes (line 8). For each $j_1 \in [m]$ that correctly computed $x_1^{j_1} = x_1$, we continue with computing $x^{j_1, j_2} = f_{j_2}(x_{\leq 0} \oplus x_1^{j_1})$ for all $j_2 \in [m]$. However, since all threads are computing the same set of tokens, we terminate all but the one corresponding to the smallest value of $j_1$ that satisfies $x_1^{j_1} = x_1$ (line 10). In essence, $C_{(m)}$ serves as a verifier, identifying drafters that miscalculated the initial part of the autoregressive computation. Once we retain one valid $j_1$, we relabel $C_{(j_1, m)}$ as the new verifier thread. We know that since $C_{(j_1)}$ returned the correct token $x_1^{j_1} = x_1$ and $x_2 = f_m(x_{\leq 1})$, the output of $C_{(j_1, m)}$ must be correct. When that thread finishes, among the remaining threads, $C_{(j_1, j_2)}$, we terminate those that miscalculated $x_2 = x_2^{j_1, m}$ (line 8) and keep only the one with $x_2^{j_1, j_2} = x_2^{j_1, m} = x_2$, whose index $j_2$ is minimal (line 10). We continue this process until the output $x^{j_1, \ldots, j_{N-1}, m}$ is obtained from the last verifier thread $C_{(j_1, \ldots, j_{N-1}, m)}$. The process of relabeling verifier threads and terminating irrelevant threads is outlined in lines 8, 10, and 11. Line 13 considers the case where the newly labeled thread may have already finished. If so, in line 14, we return to line 7 with the new verifier thread.

## D  LOOKAHEAD

We can deploy DSI on an arbitrary number of servers by selecting a sufficiently large `lookahead` hyperparameter, as explained below. Line 6 of Algorithm 1 invokes a new process to compute the target model $f_m$ immediately after generating any token (except for tokens of poisition $N$, namely, tokens corresponding to the last position). In particular, after generating a token from a drafter $f_j$ (where $j < m$). Such a "draft" token might be rejected in line 8, and is accepted otherwise, as discussed earlier. We can view this procedure as generating a draft token and sending it to "verification". Sending verification tasks of a single draft token is only a private case of DSI, where `lookahead = 1`. For a sufficiently large SP degree, the number of such verification tasks that can run in parallel is unbounded. However, in practice, the number of available processors is given, hence the SP degree must be fixed. For example, given a single drafter (that is, $m = 2$) and a single node with 8 GPUs, DSI must ensure that $SP \leq 7$, assuming that 1 GPU is sufficient for computing the drafter. To control the SP degree, we introduce a new hyperparameter, `lookahead`, which is the number of draft tokens in every verification task sent to a target server. For `lookahead > 1`, lines 2 and 6 are adjusted as follows: initiate $m - 1$ threads $C_{J \oplus (j, 1)}, C_{J \oplus (j, 2)}, \ldots, C_{J \oplus (j, m-1)}$ to generate `lookahead` tokens concurrently and respectively from $f_1, f_2, \ldots, f_{m-1}$, and initiate 1 thread $C_{J \oplus (j, m)}$ to generate 1 token from $f_m$. Here, we overload the notation for simplicity, allowing

$J \oplus (j)$ to be empty so that $J \oplus (j, j') = (j')$. This change reduces the number of invocations required. For any given models $f_1, f_2, \ldots, f_m$ and an SP degree, there exist a sufficiently large `lookahead` value such that there is at least one available target server by the time that any verification task is sent, so that verification tasks never wait to be processed by a target server. Therefore, the `lookahead` hyperparameter allows tuning DSI to use an arbitrary maximal number of available processing units.

Theoretically, without scaling the `lookahead` such that `lookahead` $\propto \frac{\text{target latency}}{\text{drafter latency}}$, the SP degree might grow to infinity. For example, if the time it takes to compute a forward pass of the drafter goes to 0 or the time it takes to compute a forward pass of the target model goes to infinity.

## E PROOFS

**Theorem 1.** *Under Assumptions 1, 2 and 3, Algorithm 1 returns the same output and runs at least as fast as running the target model itself without speculative inference (SI).*

*Proof.* We begin by demonstrating the losslessness of the algorithm. We would like to prove that when $v = k$, there is a thread $C_{J_k}$, that is the only thread that is labeled as a verifier, and it correctly computes the next token and that $J_k = J' \oplus (m)$ for some sequence $J' = (j_1, \ldots, j_{k-1})$ of length $k - 1$, where $x_i^{j_1, \ldots, j_i} = x_i$ for all $i \in [k-1]$. We will prove this by induction on the value of $v$. In addition, we note that if this pattern is appreciated by the algorithm, then it is clearly a lossless algorithm.

**Base case ($v = 1$):** Initially, when $v = 1$, there is only one verifier, $C_{(m)}$, which runs the target model $f_m$. Thus, when it finishes, it will return the correct token, $x_1$. Since the verifier is relabeled only when the value of $v$ changes (see lines 11-12), as long as $v = 1$, the only thread labeled as a verifier is $C_{(m)}$.

**Induction hypothesis:** Assume that as long as $v = k$, there is only one thread $C_{J_k}$ labeled as a verifier, which returns the correct token $x_k$, and that $J_k = J' \oplus (m)$ for some $J' = (j_1, \ldots, j_{k-1})$ of length $k - 1$, where $x_i^{j_1, \ldots, j_i} = x_i$ for all $i \in [k-1]$.

**Induction step:** When $v$ is updated from $k$ to $k + 1$, this change only occurs when the condition in line 7 is met. This condition indicates that the single verifier thread $C_{J_k}$, which is of length $|J_k| = k$, has finished computing its output token. By the induction hypothesis, this thread returns $x_k$ as its output. Since $f_m$ is slower than all drafter models $f_1, \ldots, f_{m-1}$, all threads $C_{J' \oplus (i)}$ have already finished computing their outputs. Thus, when executing lines 8, 10, and 11, the only threads that remain active are the descendants of $C_{J' \oplus (j^*)}$, and the only thread serving as a verifier is $C_{J' \oplus (j^*, m)}$. Since $x_i^{j_1, \ldots, j_i} = x_i$ for all $i \leq k - 1$ and $x_k^{j_1, \ldots, j_{k-1}, j^*} = x_k$, then $C_{J' \oplus (j^*, m)}$ simply computes the output of the target model $f_m$ on the correct sequence $x_{\leq 0} \oplus x_1 \oplus \cdots \oplus x_k$. Hence, it correctly returns the $(k+1)$th token $x_{k+1}$, as desired.

**Time:** We notice that the algorithm terminates once it has computed the output of $C_{J_N}$. By Assumption 3, we have $T_{\text{wall}} [\text{Algorithm 1}] = \sum_{i=1}^{N} T_{\text{wall}} [\text{computing } f_{j_i} (x_{\leq i})]$ and by Assumption 2, we have $T_{\text{wall}} [\text{computing } f_{j_i} (x_{\leq i})] \leq T_{\text{wall}} [\text{computing } f_m (x_{\leq i})]$. Together we obtain $T_{\text{wall}} [\text{Algorithm 1}] \leq \sum_{i=1}^{N} T_{\text{wall}} [\text{computing } f_m (x_{\leq i})]$ which is the amount of time that it takes to compute the output tokens without speculative inference. $\square$

**Proposition 1.** *Suppose we have a drafter model $f_1$, a target model $f_2$ and a prompt $x_{\leq 0}$. Assume that $f_1$ requires $t_1$ time units to compute each of its outputs, and $f_2$ requires $t_2$ time units, where $t_2 > t_1$. Assume that given the prompt $x_{\leq i} = x_{\leq 0} \oplus x_1 \oplus \cdots \oplus x_i$, the probability that $f_1$ returns the (correct) token $x_{i+1}$ is $p$. Then, the expected time it takes Algorithm 1 to calculate the correct output is at most $t_1 p(N - 1) + t_2((1 - p)(N - 1) + 1)$ time units, compared to the $t_2 N$ time units required if we were to compute $f_2$ without speculative inference.*

*Proof.* To understand how it works, let $j_1 \in \{1, 2\}$ be the smallest index such that $x_1^{j_1} = x_1$ and for all $i \in [N-1]$, we recursively define $j_i \in \{1, 2\}$ to be the smallest index such that $x_i^{j_1, \ldots, j_i} = x_i$. We also fix $j_N = 2$. In addition, let $i_0 = 0$ and $i_r$ be the $r$th index in $[N]$ such that $j_{i_r} = 2$. We notice that it takes $t_1(i_1 - 1) + t_2$ time units to compute the value of $x_{i_1}^{j_1, \ldots, j_{i_1}}$. This is because we first compute $x_1^1$,

then $x_1^{1,1}$, continuing up to $x_{i_1-1}^{1,\ldots,1}$, and finally $x_{i_1}^{1,\ldots,1,2}$. Each of the first $(i_1 - 1)$ tokens takes $t_1$ time units, while the final token takes $t_2$ time units. After $t_1(i_1-1)+t_2$ time units, we will have computed $x_1^2, x_2^{1,2}, x_3^{1,1,2}$, and so on, up to $x_{i_1}^{1,\ldots,1,2}$. Since $f_1$ consistently generates accurate tokens up to index $i_1-1$, once we observe that $x_1^2$ matches $x_1^1$, we know that $x_2^{1,2} = x_2$ and can then verify that $x_2^{1,1} = x_2$ is also correct. Once we verify that $x_2^{1,1} = x_2$, we can verify $x_2^{1,1,2}$ and continue this pattern to verify $x_2^{1,1,1}$, and so forth. We note that calculating all of these tokens up to the calculation of $x_{i_1}^{1,\ldots,1,2}$ take at most $t_1(i_1-1)+t_2$ time units. Thus, we can verify that $x_{i_1}^{1,\ldots,1,2} = x_{i_1}$ with at most $t_1(i_1-1)+t_2$ time units. By the same argument as above, it takes $\sum_r (t_1((i_r - i_{r-1}) - 1) + t_2)$ time units to compute the value of $x_N^{j_1,\ldots,j_N}$ (and to verify its correctness). We notice that $Q = \sum_r (i_r - i_{r-1} - 1)$ is the number of indices $i \in [N-1]$ such that $j_i = 1$. Since $\mathbb{E}[Q] = p(N-1)$, we have $\mathbb{E}\left[\sum_r (t_1((i_r - i_{r-1}) - 1) + t_2)\right] = t_1 p(N-1) + t_2((1-p)(N-1)+1)$. $\square$

**Theorem 2.** *Under Assumptions 1, 2 and 3, Algorithm 1 runs at least as fast as SI in expectation.*

*Proof.* Suppose we have a drafter model $f_1$, a target model $f_2$, and a prompt $x_{\le 0}$. Assume that $f_1$ requires $t_1$ time units to compute each of its outputs, and $f_2$ requires $t_2$ time units, where $t_1 < t_2$. Assume that given the prompt $x_{\le i} = x_{\le 0} \oplus x_1 \oplus \cdots \oplus x_i$, the probability that $f_1$ returns the (correct) token $x_{i+1}$ is $p$. Consider generating $N > k + 1$ tokens from $f_2$ using the SI (or DSI) algorithm with $\texttt{lookahead} = k$. At time $= 0$, SI starts generating draft tokens, by the definition of SI. At time $= kt_1$, SI completes generating the first $k$ draft tokens $x_1^1, x_2^{1,1}, \ldots, x_k^{1,\ldots,1}$. At time $= kt_1 + t_2$, SI completes verifying the first $k$ tokens $x_1^1, x_2^{1,1}, \ldots, x_k^{1,\ldots,1}$. Let $A_1, A_2, \ldots, A_{k+1}$ be indicator variables sampled as follows. $A_i = 1$ with probability $p$ and $A_i = 0$ otherwise, for all $i \in [k+1]$. Let $n := \min\{i | A_i = 0\} - 1$. Note that $n$ is distributed as the number of accepted drafts among the first $k$ drafts of SI (or DSI). SI completes generating the first $n+1$ tokens at time $= kt_1 + t_2$ for any $n \in \{0, 1, \ldots, k\}$, by the definition of SI. The first iteration of SI cannot output tokens at positions $> k + 1$, by the definition of SI. The earliest time at which SI can complete generating $x_{k+2}$ is by the end of its second iteration. Hence, SI completes generating $x_{k+2}$ at time $\ge 2(kt_1 + t_2)$. Consider DSI with the same $f_1$, $f_2$, and $\texttt{lookahead}$ over at least $\left\lceil \frac{t_2}{kt_1} \right\rceil$ servers. We show that DSI can complete generating $x_{k+2}$ at time $\le 2(kt_1 + t_2)$, and, in expectation, at time $< 2(kt_1 + t_2)$. By the definition of DSI, DSI never preempt the current verifier. At time $= kt_1$, DSI invokes concurrently (i) the verifying of the batch containing the first $k$ tokens $x_1^1, x_2^{1,1}, \ldots, x_k^{1,\ldots,1}$ that are not yet verified, and (ii) the drafting of $x_{k+1}^{1,\ldots,1} x_{k+2}^{1,\ldots,1}, \ldots, x_{2k}^{1,\ldots,1}$, by the definition of DSI. We use a coupling argument to align the two algorithms over the indicator variables $A_i$ for all $i$. If $n = k$, then both SI and DSI complete generating the $(k+1)$th first token $x_{k+1}$ at time $= kt_1 + t_2$. At that time, DSI either invokes a new current verifier thread or labels an existing thread as the current verifier (depending on $\frac{t_2}{t_1}$ and the $\texttt{lookahead}$). Hence, DSI completes generating $x_{k+2}$ at time $\le kt_1 + 2t_2$, exactly when the current verifier thread completes its verification. DSI is faster than SI for all $t_1, t_2, k$ since $kt_1 + 2t_2 < 2(kt_1 + t_2)$. Otherwise, both algorithms accept the first $n+1$ tokens at time $\le kt_1 + t_2$. At that time, the proof repeats for $N - (n+1)$. $\square$

## F  EXPERIMENTS DETAILS

### F.1  TTFT AND TPOT

This section elaborates on our separate experiment that estimates the expected TTFT and TPOT.

To ensure the $\texttt{wait}$ waiting times in our simulations are realistic, we distinguish between time to first token (TTFT) and time per output token (TPOT). We estimated the TTFT and TPOT latencies for each combination of a model and a dataset independently on a single NVIDIA A100 80GB GPU.

For each combination $\langle d, f \rangle$ of a dataset $d$ and a corresponding target or drafter model $f$, we estimate the average latencies of $f$ in the following manner. First, we select 50 prompts from $d$ uniformly at random, and for each prompt, generate 20 tokens using $f$, measuring the latency for each token in milliseconds. Following prior work, we distinguish between Time to First Token (TTFT) generation and Time Per Output Token (TPOT) generation (of all subsequent 19 tokens). Finally, we calculate the average TTFTs and TPOTs over all prompts per model-dataset pair, to estimate the expected latency

of a single forward pass. In all our experiments that measure the latency of computing forwards (i.e., TTFT and TPOT), models run with an MP degree of one and without memory offloading. We used a single NVIDIA A100 80GB GPU and the measured model was fully loaded to the GPU memory.

Our main experiment (Table 2) and an ablation experiment (§4.1) both use the estimated TTFT and TPOT as follows: generating the first token adds a `wait` of TTFT while generating each subsequent token adds a `wait` of TPOT. Since TTFT is usually significantly longer than TPOT (which dominates the overall sequence generation time), all latency figures in Table 2 refer to TPOT, for brevity. The estimated TPOT latency of the target model and the drafter are shown in "Target Latency (ms)" and "Drafter Latency (ms)", respectively. We also report the ratio between the target and drafter latencies and present it in percentages ("Drafter Latency (%)"). The effective prefilling-decoding latencies ratio of every pair of a model and a dataset is provided in Table 3.

Table 3: The ratio between the estimated time to first token (TTFT) and time per output token (TPOT) for various off-the-shelf models and datasets.

| Model | Dataset | TTFT/TPOT Ratio |
| --- | --- | --- |
| lmsys/vicuna-13b-v1.3 | cnn_dailymail | 5.36 |
| double7/vicuna-68m | cnn_dailymail | 1.04 |
| lmsys/vicuna-13b-v1.3 | danielkorat/alpaca | 1.15 |
| double7/vicuna-68m | danielkorat/alpaca | 1.05 |
| lmsys/vicuna-7b-v1.3 | cnn_dailymail | 4.53 |
| double7/vicuna-68m | cnn_dailymail | 1.06 |
| lmsys/vicuna-7b-v1.3 | danielkorat/alpaca | 1.19 |
| double7/vicuna-68m | danielkorat/alpaca | 1.06 |
| bigcode/starcoder | openai/openai_humaneval | 1.35 |
| bigcode/tiny_starcoder_py | openai/openai_humaneval | 1.19 |
| bigcode/starcoder | mbpp | 1.54 |
| bigcode/tiny_starcoder_py | mbpp | 1.20 |
| microsoft/Phi-3-medium-128k-instruct | openai/openai_humaneval | 1.29 |
| microsoft/Phi-3-mini-128k-instruct | openai/openai_humaneval | 1.23 |
| microsoft/Phi-3-medium-128k-instruct | mbpp | 1.43 |
| microsoft/Phi-3-mini-128k-instruct | mbpp | 1.27 |
| microsoft/Phi-3-medium-128k-instruct | cnn_dailymail | 4.77 |
| microsoft/Phi-3-mini-128k-instruct | cnn_dailymail | 3.88 |

## F.2 ACCEPTANCE RATE

This section elaborates on our separate experiment that estimates the expected acceptance rate.

To ensure our evaluation is realistic, we used real-world acceptance rates, calculated as follows. For any given combination of ⟨target, drafter, dataset⟩, we estimate their acceptance rate independently. For each input prompt from the dataset, we generate tokens from both the drafter and the target model. We then consider the lengths of the longest sequences of exact token matches between the target and the drafter. Below is a simplified example where tokens are counted as English words. If the target generates "We can only see a short distance ahead, but we can see plenty there that needs to be done. [...]" and the drafter generates "We can only see a short distance ahead, we done. [...]", then the longest sequence of exact matches is 8 tokens long. The expected number of accepted drafts is $\bar{n} := \frac{1}{N} \sum_{i=1}^{N} n_i$ where $n_i$ is the number of accepted draft tokens for the $i$th prompt. The acceptance rate is then calculated from a geometric distribution, (acceptance rate) $:= 1 - \frac{1}{1+\bar{n}}$. In general, the reported acceptance rate is guaranteed to converge to the expected acceptance rate as $N \to \infty$ (Appendix F.2.1). In this experiment, the reported acceptance rate is calculated based on generating 256 tokens for each prompt.

This experiment suggests that off-the-shelf model "families" like StarCoder (Li et al., 2023a) or Vicuna (Zheng et al., 2023) can form good pairs of target and drafter because their acceptance rates are relatively higher, as reported in Table 2. These families consist of models of different sizes that were trained similarly and on similar datasets. We observe that even relatively small drafters demonstrate good alignment with larger models from the same family. For example, `Starcoder-168M` (drafter) and `Starcoder-15B` (target) yield an acceptance rate of 93%.

### F.2.1 ACCEPTANCE RATE FROM GEOMETRIC DISTRIBUTION

Both our main experiment (Table 2) and ablation (§4.1) measure the total latency of Non-SI, SI, and DSI. The total latency is dominated by the target and drafter forward passes. The numbers of required target and drafter forwards depend on the acceptance rate. To count the number of forwards, we can simulate the algorithms by sampling from the models, execute the verification procedure (for SI and DSI), and count the number of required forwards. Since sampling from models is inherently stochastic, the total latency $T$ is a random variable, and we focus on its expectation, $\mathbb{E}[T]$.

There exists an acceptance rate $p$ such that the expected latency satisfies $\mathbb{E}[T] = T$. This acceptance rate can be estimated in two ways. The first method directly computes the likelihood that a token is accepted by the verifier. Alternatively, $p$ can be estimated by fitting a geometric distribution to the acceptance process. Specifically, this involves calculating the average number of tokens accepted per iteration and extracting $p$ as the parameter of the geometric distribution.

Both approaches are consistent. As the number of iterations approaches infinity, the estimated acceptance rate converges to the true empirical acceptance rate $p$. Consequently, the estimated total latency based on $p$ converges to the actual total latency. This guarantees that the reported latency and the computed latency are equal in expectation.

Prior works in the field have developed theories and methods based on the simplifying assumption that token acceptance is independently and identically distributed (i.i.d.); for example, see Leviathan et al. (2023). In this paper, estimating the acceptance rate from the fitted geometric distribution is based on the same assumption. The results in Mamou et al. (2024) reveal that such an assumption is actually realistic. Figure 4 in Mamou et al. (2024) shows that the number of accepted tokens per iteration follows a geometric distribution.

### F.3 ABLATION VIA OFFLINE SIMULATION

This section provides more implementation details about the ablation analysis, involving a complementary "offline" experiment (§4.1).

We simulate non-SI over all possible configurations of ⟨drafter latency, acceptance rate⟩ which are the cartesian product $\{0.01, 0.02, \ldots, 1\} \times \{0, 0.01, 0.02, \ldots, 1\}$ respectively. We simulate SI over all possible configurations of ⟨drafter latency, acceptance rate, `lookahead`⟩ where `lookahead` $\in \{1, 2, \ldots, 200\}$. However, given an arbitrary number of available servers, it is possible to tune the `lookahead` hyperparameter such that DSI orchestrates only available servers. To ensure that DSI could be deployed on a single node with up to eight GPUs, we simulate DSI over all configurations of SI that satisfy Equation 1 for SP $= 7$, assuming that the drafter runs on a single GPU. For each combination of ⟨drafter latency, acceptance rate, `lookahead`⟩, we run the algorithm (SI or DSI) for five repeats and average the results. Since the `lookahead` is a tunable parameter, our experiment assumes that it will be optimized by the user so that SI is optimized for each configuration. To let SI select its optimal `lookahead`, the expected latency for each ⟨drafter latency, acceptance rate⟩ configuration is the minimal average latency among all the `lookahead` values.

To calculate the speedup of algorithm X over algorithm Y per ⟨drafter latency, acceptance rate⟩, we divide the latency of Y by the latency of X. The speedups are not smooth for drafter latencies $< 20\%$ due to the discretization of the `lookahead` hyperparameter. For example, the speedups for `lookahead` $= 5$ are smooth for both SI and DSI as seen in Appendix F.7.

As in online experiments, every forward pass in offline experiments is replaced with adding the realistic `wait` time to the total latency. Therefore, the latency of non-SI in offline simulations is simply the target forward latency (i.e., the average time that it takes to compute a single forward pass of the target model, which is measured in a separate experiment that is described in the previous section) times the number of tokens to generate. For example, if the target forward latency is 30ms and the number of tokens to generate is 100, then the latency of non-SI is 3000ms. To estimate the expected latency of SI, we must introduce the `lookahead` hyperparameter and the acceptance rate of the drafter. Under the randomness of the acceptance rate, we count the number of target and drafter forward passes. Every draft token that is accepted is equivalent to generating another output token, hence reducing the number of remaining target forwards by one. For example, if the acceptance rate corresponds to accepting 1.5 drafts per iteration, then the expected number of target

forward passes is $\left\lceil \frac{100}{2.5} \right\rceil = 40$ because every target forward is expected to yield $1.5 + 1 = 2.5$ output tokens (i.e., accepted drafts plus an additional token). The number of drafter forwards depends on the `lookahead` hyperparameter. For example, if `lookahead` $= 5$, then the number of drafter forwards is $40 \cdot 5 = 200$, because we have 5 drafter forwards for each target forward. The expected latency of SI is then the sum of the expected latencies of the target and drafter forwards. For example, if the drafter forward latency is 6ms, then the expected latency of SI is $200 \cdot 6 + 40 \cdot 30 = 2400$ms, which is 1.25x faster than non-SI. The simulation of DSI is similar to the simulation of SI. In both, the only randomness is the acceptance rate of the drafter. However, since DSI can successfully hide the latency of almost all the target forwards, naive summing of the forward pass latencies cannot estimate its total latency accurately. Instead, the DSI simulation is based on an analysis that sums target forward latencies only if they are not hidden by the speculation parallelism.

### F.4 SIMULATION OF SI

We open-sourced our implementation of DSI for the single-node setup and the code for all the experiments in this paper. Below is the code for estimating an lower bound of the end-to-end latency of the SI algorithm. The end-to-end latency is the overall wall time, including the prefilling and decoding latency, and excluding any tokenization latency. It is an lower bound because we ignore any real-world latencies differ from the forward passes latencies. For example, we ignore all the latencies corresponding to switching between the models, communication between the CPU and the GPU, etc.

```python
def si(target_latency: float, drafter_latency: float, lookahead: int, N: int) -> float:
    total_cost: float = 0
    total_toks: int = 0
    while total_toks < N:
        num_accepted: int = get_num_accepted()
        total_toks += num_accepted + 1
        total_cost += lookahead * drafter_latency + target_latency
    return total_cost
```

### F.5 MODELS

For all models, we retrieve model weights from Hugging Face. For clarity and reproducibility, we provide the URLs for each model used:

- `Vicuna-13B:` `https://huggingface.co/lmsys/vicuna-13b-v1.3,` distributed under Non-Commercial License.
- `Vicuna-7B:` `https://huggingface.co/lmsys/vicuna-7b-v1.3,` distributed under Non-Commercial License.
- `Vicuna-68M:` `https://huggingface.co/double7/vicuna-68m`, distributed under the Apache License 2.0.
- `Starcoder-15B:` `https://huggingface.co/bigcode/starcoder,` distributed under the Responsible AI License.
- `Starcoder-168M:` `https://huggingface.co/bigcode/tiny_starcoder_py`, also distributed under the Responsible AI License.
- `Phi3-14B:` `https://huggingface.co/microsoft/Phi-3-medium-128k-instruct` distributed under the MIT license.
- `Phi3-4B:` `https://huggingface.co/microsoft/Phi-3-mini-128k-instruct` distributed under the MIT license.

### F.6 DATASETS AND PROMPTS

We use standard datasets from Hugging Face and standard prompts from the state-of-the-art.

#### F.6.1 MBPP

MBPP dataset consists of crowd-sourced Python programming problems and is distributed under the cc-by-4.0 License.

Concerning the prompt, we followed (Ben Allal et al., 2022; Fried et al., 2023) and included the description of the programming task and a single test to verify solution, in order to help the model catch the signature of the function (see Figure 3).

```
"""{text}
{test_list[0]}
"""
```

Figure 3: MBPP Prompt

### F.6.2 HUMANEVAL

HumanEval dataset includes programming problems and is distributed under the MIT License.

Prompt contains only `prompt` field from the dataset.

### F.6.3 CNN-DM

CNN-DM contains news articles and is distributed under the Apache License 2.0.

We included the `article` field in the prompt as in Figure 4.

```
"""Summarize:
{article}
Summary:
"""
```

Figure 4: CNN-DM Prompt

### F.6.4 ALPACA

Alpaca dataset contains instructions and demonstrations. It is distributed under the cc-by-nc-4.0 License.

We follow Taori et al. (2023) to define the prompts. For samples with a non-empty input field, we use the prompt as in Figure 5 while for samples with empty input field, we use the prompt as in Figure 6.

```
"""Below is an instruction that describes a
task, paired with an input that provides
further context. Write a response that
appropriately completes the request.

### Instruction:
{instruction}

### Input:
{input}

### Response:
"""
```

Figure 5: Alpaca prompt for samples with a non-empty input field.

```
"""Below is an instruction that describes a
task. Write a response that appropriately
completes the request.

### Instruction:
{instruction}

### Response:
"""
```

Figure 6: Alpaca prompt for samples with empty input field.

### F.7 SPEEDUPS FOR LOOKAHEAD = 5

Figure 7 illustrates the performance comparison between Speculative Inference (SI), non-Speculative Inference (non-SI), and Dynamic Speculative Inference (DSI) for a static lookahead of $\texttt{lookahead} = 5$. The figure consists of three heatmaps, each representing a pairwise comparison of these algorithms across different drafter speeds and accuracies.

In Figure 7(a), we compare SI to non-SI. The pink regions indicate scenarios where SI is slower than non-SI, which occurs when the drafter is either too slow or inaccurate. This highlights the limitations of SI in certain conditions.

Figures 7(b) and (c) demonstrate that DSI consistently outperforms both SI and non-SI across all drafter configurations. This empirical evidence supports our theoretical findings that DSI is faster than both SI and non-SI in expectation, regardless of the drafter's speed or accuracy.

These results underscore the robustness and efficiency of DSI compared to existing inference methods, particularly in scenarios where SI might falter due to suboptimal drafter performance.

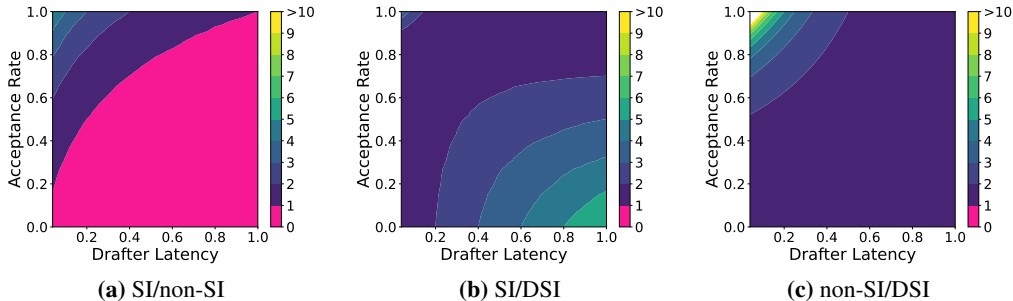

    **(a)** SI/non-SI         **(b)** SI/DSI         **(c)** non-SI/DSI

Figure 7: Each heatmap (labeled "X/Y") plots the ratio between the run time of algorithm X and the run time of algorithm Y. SI is run with $\texttt{lookahead} = 5$. **(a)**: SI is slower than non-speculative inference (non-SI) when the drafter is either slow or inaccurate enough (pink marks slowdowns). DSI is never slower than either SI or non-SI. **(b, c)**: DSI is always faster than speculative inference (SI) and non-speculative inference (non-SI) algorithms for various drafters.

