# OpenReview forum: "Distributed Speculative Inference (DSI): Speculation Parallelism for Provably Faster Lossless Language Model Inference"
_ICLR.cc/2025/Conference — ICLR 2025 Poster_

### Official Review · Reviewer_LmXh · 2024-11-03

**Soundness:** 2
**Presentation:** 3
**Contribution:** 2
**Rating:** 5
**Confidence:** 3

**Summary:**

This work proposes distributed speculative inference (DSI), or speculation parallelism in particular,
which accelerates speculative inference (SI) by leveraging more hardware resources (e.g. GPUs) to hide the latency of the verification steps in SI via parallelism and careful synchronization across multiple threads.
Through rigorous analysis,
it is shown that, while SI can be slower than standard autoregressive inference (non-SI) in certain scenarios,
DSI is provably faster in expectation than both of them.
With a proof-of-concept implementation of DSI and careful design of numerical simulations,
empirical results suggest that DSI achieves 1.29-1.92x speedup over SI in various settings.

**Strengths:**

- The proposed speculation parallelism is novel and interesting, and might inspire future work on speculative inference or more general LLM inference acceleration.

- The rigor in both theoretical analysis and numerical validation (via careful design of simulations) is appreciated.

- The source code for the experiments is provided.

**Weaknesses:**

My main concern is that the practical value of this work is not yet evident enough.
If I understand correctly, the implementation of DSI in the current work is more or less a proof-of-concept, rather than one that is ready for application.
More specifically, only the outline of the multi-thread system is implemented, while the LLM part (e.g. forward passes of the drafter or target model) is not actually implemented but rather replaced by `WAIT` commands with delays that mimic the actual latencies of real LLM forward passes, for the purpose of simulations.
One might argue that, for a full implementation of DSI, we simply need to plug the LLM forward passes into the multi-thread system.
But this does not seem straightforward to me, since LLM inference is a complicated process by itself;
combining it with the implemented multi-thread system can be challenging, as these two might not be completely decoupled.



One particular aspect I can think of is the compatibility with KV caches, which have been supported by both non-SI and SI, and enabled by default in most cases (e.g. when huggingface transformers is used for profiling the latency of standard LLM inference in the `ExperimentLatency` class within the source code of this work).
With multiple target LLMs (or verifiers) on different GPUs, each at its own pace,
managing and synchronizing their KV caches seems like a non-trivial task and requires quite some engineering work,
which has not been implemented in this work.

**Questions:**

- Do you think DSI with a batch size larger than 1 is possible, at least in theory?
Batch inference is already quite a challenge for SI, and things seem to get much more complicated with the proposed speculation parallelism.


- Some minor issues about writing:
  - The paragraph between Lines 422 - 487 is a bit long; might be better to divide it into several paragraphs
  - Line 128 mentions "implementing (3) above", while Eq. (3) is actually located in the appendix
  - Line 303, $f_1$ and $f_2$: should they be $f_j$ and $f_m$ instead?
  - Line 931, "upper bound": should it be "lower bound", since some components of latency in reality are neglected in the estimate?

---

> ### Author Response · Authors · 2024-11-23
>
> **Response to Reviewer LmXh**
>
> We sincerely thank you for your detailed and thoughtful review. We are glad that you found our proposed speculation parallelism novel and interesting and that you appreciated the rigor in our theoretical analysis and simulations.
>
> **Acknowledged Strengths:**
>
> - **Novelty and Inspiration:** We appreciate your recognition that our proposed speculation parallelism is novel and could inspire future work on speculative inference and LLM inference acceleration.
>
> - **Rigor in Analysis and Validation:** Thank you for highlighting the rigor in both our theoretical analysis and the careful design of numerical simulations validating our approach.
>
> - **Transparency:** We are pleased that you appreciated our provision of source code for the experiments, which demonstrates our commitment to transparency and reproducibility.
>
> **Responses to Weaknesses and Questions:**
>
> > **Weakness:** My main concern is that the practical value of this work is not yet evident enough. If I understand correctly, the implementation of DSI in the current work is more or less a proof-of-concept, rather than one that is ready for application. More specifically, only the outline of the multi-thread system is implemented, while the LLM part (e.g., forward passes of the drafter or target model) is not actually implemented but rather replaced by WAIT commands with delays that mimic the actual latencies of real LLM forward passes, for the purpose of simulations.
>
> **Response:** Your understanding of our proof-of-concept is correct. What we have are theoretical guarantees backed by simulations. The simulations show the expected speedups for well-known models (Table 2) and for any possible drafter (Figure 2) in the single-node setup.
>
> ---
>
> > **Weakness:** Combining LLM inference with the implemented multi-thread system can be challenging, as these two might not be completely decoupled. One particular aspect I can think of is the compatibility with KV caches, which have been supported by both non-SI and SI, and enabled by default in most cases. With multiple target LLMs (or verifiers) on different GPUs, each at its own pace, managing and synchronizing their KV caches seems like a non-trivial task and requires quite some engineering work, which has not been implemented in this work.
>
> **Response:** DSI essentially constructs and verifies token trees on the fly. Efficient KV cache management of token trees has already been developed in SpecInfer, where tree paths can share common prefixes. Practitioners who have access to a node with 8 GPUs can apply SpecInfer’s KV cache management as-is to achieve the expected speedups reported in this paper. While it might require some engineering effort to implement SpecInfer’s KV cache management, it is a solved research problem and has been shown to add negligible latency. Therefore, the orchestration (DSI) is essentially decoupled from the underlying computation of forwards (including KV cache management) not only theoretically but also practically.
>
> *Miao, Xupeng, et al. "SpecInfer: Accelerating Generative Large Language Model Serving with Tree-based Speculative Inference and Verification." arXiv preprint arXiv:2305.09781 (2023).*
>
> ---
>
> > **Question:** Do you think DSI with a batch size larger than 1 is possible, at least in theory? Batch inference is already quite a challenge for SI, and things seem to get much more complicated with the proposed speculation parallelism.
>
> **Response:** Great question, and a very exciting future direction with huge potential! It has recently been shown that SI is effective in the multi-request setup (where the batch size is often larger than one) for sufficiently long sequences [^1]. No fundamental limitations prevent practitioners from applying DSI to multi-request setups. This paper covers all the foundational concepts of such an implementation of DSI. However, we believe that analyzing or simulating the expected speedups is not straightforward. Unfortunately, we do not currently have access to a node with 8 GPUs to further research this direction.
>
> [^1]: https://www.together.ai/blog/speculative-decoding-for-high-throughput-long-context-inference
>
> ---
>
> > **Minor Writing Issues:**
> >
> > - The paragraph between Lines 422–487 is a bit long; might be better to divide it into several paragraphs.
> > - Line 128 mentions "implementing (3) above," while Eq. (3) is actually located in the appendix.
> > - Line 303: Symbols
> *f1* and
> *f2* should be
> *fj* and
> *fm* instead.
> > - Line 931: "upper bound" should be "lower bound," since some components of latency in reality are neglected in the estimate.
>
> **Response:** Thank you for carefully catching these typos. We will revise the paper to address all these issues and improve the clarity of the paragraph between Lines 422–487.
>
> ---
>
> We appreciate your insightful comments and constructive feedback, which have already helped us improve the paper. Thank you again for your thoughtful review.

---

> ### Comment · Reviewer_LmXh · 2024-12-02
>
> I'd like to thank the authors for their responses. My overall perspective on this work hasn't changed much, thus I'm inclined to keep my rating for now.

---

### Official Review · Reviewer_4xNe · 2024-11-04

**Soundness:** 3
**Presentation:** 3
**Contribution:** 3
**Rating:** 6
**Confidence:** 3

**Summary:**

The paper introduces the parallel verification for accelerating speculative decoding. Experiments show speedups compared to the original speculative decoding.

**Strengths:**

- The paper is well-written and easy to follow.

- The idea is clear and seems to work well through evaluation.

**Weaknesses:**

- The largest target model is capped at 13B.

  - How does the model work on larger models?

- The method need multiple GPUs to host several target model. This can cause deployment issues and energy issues.

  - How does the data communication defect the approach? The KV-cache also seems to move between GPUs. How about synchronization issues?
  - The analysis of the energy consumption compared to the original speculative decoding method seems to missing. More justification may be needed given the energy inefficiency of the method.

**Questions:**

See above.

---

> ### Author Response · Authors · 2024-11-23
>
> We sincerely thank you for your thoughtful review and valuable feedback. We are glad that you found the paper well-written and easy to follow and appreciated the clarity and effectiveness of our proposed method.
>
> **Acknowledged Strengths:**
>
> - **Clarity and Readability:** We appreciate your positive remarks regarding the clarity and ease of understanding of the paper.
>
> - **Effectiveness of the Idea:** We are pleased that you found our proposed idea clear and supported by the evaluation results.
>
> **Responses to Weaknesses and Questions:**
>
> > **Weakness:** The largest target model is capped at 13B. How does the model work on larger models?
>
> **Response:** The largest target model we evaluate is 15B (StarCoder). The guarantees of DSI hold for any model size, with the expected speedup provided in Figure 2. As the target model size increases while the drafter remains fixed, the "drafter latency" becomes smaller because it represents the ratio between the latencies of the two models.
>
> ---
>
> > **Weakness:** How does the data communication defect the approach? The KV-cache also seems to move between GPUs. How about synchronization issues?
>
> **Response:** Each server maintains its own KV cache. The servers collaboratively process a token tree with shared prefixes. Therefore, the implementation directly follows the token tree KV cache management method of SpecInfer. Synchronizations occur at every draft rejection.
>
> *Miao, Xupeng, et al. "SpecInfer: Accelerating Generative Large Language Model Serving with Tree-based Speculative Inference and Verification." arXiv preprint arXiv:2305.09781 (2023).*
>
> ---
>
> > **Weakness:** The analysis of the energy consumption compared to the original speculative decoding method seems to be missing. More justification may be needed given the energy inefficiency of the method.
>
> **Response:** How do you define “energy consumption” in this context, and what analysis do you believe is missing?
>
> ---
>
> We hope that our responses address your concerns and provide clarity. Thank you once again for your time and consideration. We look forward to any further suggestions you might have.

---

### Official Review · Reviewer_kPPx · 2024-11-07

**Soundness:** 2
**Presentation:** 2
**Contribution:** 2
**Rating:** 3
**Confidence:** 4

**Summary:**

The paper introduces Distributed Speculative Inference (DSI), an inference algorithm that accelerates large language model (LLM) inference more effectively than traditional Speculative Inference (SI) methods or standard autoregressive techniques. Unlike SI, DSI achieves speed improvements even when these drafters are slow or less precise. Speculation Parallelism (SP) enables target and drafter models to overlap in execution, reducing latency and introducing a new balance between computational resources and response time

**Strengths:**

* Paper is well written upto method.
* Proposed Speculation parallelism effectively solves the blocking operation of speculative inference into non-blocking.
* Method is backed by proof.

**Weaknesses:**

* Evaluations are very weak. Can we get the experimental result by varying lookahead values on different machines?
* Actual improvement measurement seems missing.
* Evaluating distributed evaluation on a single machine doesn’t make sense.. Authors really need to find way to run the evaluation on multi-GPU machines. Distribution involves communication latency as well, so in practice there are many things involved but these practical concerns are not discussed in the paper.
* Evaluation writing is also very confusing..

**Questions:**

* What kind of communications are involved in distributed speculative parallelism and how does that affect the latency?

---

> ### Author Response · Authors · 2024-11-21
>
> We sincerely thank you for your thoughtful review and valuable feedback. We are pleased that you recognize the strengths of our approach, particularly the effectiveness of Speculation Parallelism in transforming the blocking operation of Speculative Inference (SI) into a non-blocking one, and that our method is backed by proof.
>
> **Acknowledged Strengths:**
>
> - **Clarity up to the Method Section:** We appreciate your positive comment that the paper is well-written up to the method section.
>
> - **Effectiveness of Speculation Parallelism:** We're glad that you acknowledge how our proposed Speculation Parallelism effectively solves the blocking operation of SI, making it non-blocking.
>
> - **Method Backed by Proof:** Your recognition that our method is supported by theoretical proofs is encouraging.
>
> **Responses to Weaknesses and Questions:**
>
> > **Weakness:** Evaluations are very weak. Can we get the experimental result by varying lookahead values on different machines?
>
> **Response:** The reported results are based on simulations. They rely on estimated TTFT and TPOT—as measured “offline,” namely, before running the simulations. Changing machines will likely change the TTFT and TPOT, and therefore the effective “drafter latency.” Figure 2 provides the expected speedups for any drafter latency, and therefore covers all possible single-node machine configurations with 8 GPUs.
>
> ---
>
> > **Weakness:** Actual improvement measurement seems missing.
>
> **Response:** Could you please clarify what specific measurements you believe are missing? The actual improvement measurements are reported in our results section, demonstrating the speedups achieved by DSI over SI and standard methods.
>
> ---
>
> > **Weakness:** Evaluating distributed evaluation on a single machine doesn’t make sense. Authors really need to find a way to run the evaluation on multi-GPU machines. Distribution involves communication latency as well, so in practice there are many things involved, but these practical concerns are not discussed in the paper.
>
> **Response:** The theoretical guarantees of DSI hold for both single- and multi-node setups. Unfortunately, we do not have the resources to evaluate a complete implementation of DSI over even a single node with 8 GPUs. Consequently, evaluating a multi-node implementation of DSI (orchestrating more than 8 GPUs) is not planned.
>
> ---
>
> > **Weakness:** Evaluation writing is also very confusing.
>
> **Response:** We will rewrite this part to improve clarity and ensure that the evaluation methodology and results are clearly communicated.
>
> ---
>
> > **Question:** What kind of communications are involved in distributed speculative parallelism and how does that affect the latency?
>
> **Response:** The speedups reported in Table 2 are based on simulations where DSI is implemented as a multithreaded system, and therefore incur all the real-world latencies of such systems, including context switching.
>
> ---
>
> We hope that our responses address your concerns and provide the necessary clarifications. We will work on improving the evaluation section to enhance clarity and address the issues you've raised. Thank you once again for your time and consideration.

---

### Official Review · Reviewer_U6hk · 2024-11-09

**Soundness:** 3
**Presentation:** 3
**Contribution:** 3
**Rating:** 6
**Confidence:** 3

**Summary:**

The paper introduces a novel approach called Distributed Speculative Inference (DSI) for accelerating LLM inference. The algorithm is inspired by Speculative Inference (SI) but uses multiple drafters (small) and overlaps their execution with verifications from the verifier (large) to obtain greater speedup than SI. The authors show theoretically that DSI is faster than SI and standard LLM inference in expectation and
 also show empirical speedups.

**Strengths:**

1. DSI appears to be a novel approach and effectively leverages the strengths of SI but also seeks to address its weaknesses like the drafter being blocked by verifier until the generated tokens are verified and also generalizes SI to work with multiple drafters in parallel.

2. The paper provides empirical evidence demonstrating that DSI outperforms both traditional speculative inference (SI) and non-SI methods

**Weaknesses:**

1. The approach requires more compute and memory (multiple KV cache) than SI with the amount of extra compute and memory required increasing with increase in number and size of the drafters, and with increase in input context length. Therefore, I believe that the increase in cost/resources also needs to also be considered in addition to the reduction in latency to make a fair assessment.

2. Evaluation is performed using simulations and not real implementations due to which it is not entirely clear if the gains will hold up in real settings. However, I understand that it is challenging to procure the amount of compute for inference with multiple LLMs and so I am willing to let this pass if my questions below on the evaluation are adequately answered.

**Questions:**

1. I do not understand the rationale behind the formula for acceptance rate on Page 8, line 410. Please clarify.

2. Why are you using an estimate of the acceptance rate instead of independently simulating acceptance/rejection for each drafter, target pair? For e.g. you could generate 'lookahead' tokens, verify them, go back to the drafter, and so on until the entire sequence is generated and then use the estimates of TTFT and TPOT to estimate the total latency.

3. Please report the value of lookahead used for each row of Table 2, and clarify why that value was chosen.

4. Why are real world latencies of multithreading and context switching ignored (Page 10, lines 487-488)? These contribute to the overhead of DSI, and I feel that they should be reported and studied for a fair comparison of with SI and non-SI.

5. Will it be possible to show any simulation results with multiple drafters? I feel that one of the strengths of DSI is that it can work with multiple drafters and I would like to see how using multiple drafters compares with using a single drafter.

---

> ### Author Response · Authors · 2024-11-21
>
> We sincerely thank you for your thoughtful review and valuable feedback. We are pleased that you recognize the **novelty** of our approach and its **effectiveness** in addressing the fundamental limitation of Speculative Inference (SI) as a sequential algorithm.
>
> **Acknowledged Strengths:**
>
> - **Novelty of DSI:** We appreciate your acknowledgment that Distributed Speculative Inference (DSI) is a novel approach that not only builds upon SI but also addresses its weaknesses, such as the drafter being blocked by the verifier until generated tokens are verified. Your recognition of DSI's ability to generalize SI to work with multiple drafters in parallel is encouraging.
>
> - **Empirical Evidence of Performance:** We're glad that you find our empirical evidence compelling in demonstrating that DSI outperforms both traditional SI and non-SI methods. Your positive remarks on our experimental results motivate us to further our research in this area.
>
> **Responses to Weaknesses and Questions:**
>
> > Weakness: The approach requires more compute and memory (multiple KV caches) than SI, with the amount of extra compute and memory required increasing with the number and size of the drafters, and with an increase in input context length. Therefore, I believe that the increase in cost/resources also needs to be considered in addition to the reduction in latency to make a fair assessment.
>
> Lines 273-291 compare DSI, tensor parallelism (TP), and pipeline parallelism (PP) over the same budget.
>
> ---
>
> > Question 1: I do not understand the rationale behind the formula for acceptance rate on Page 8, line 410. Please clarify.
>
> We estimate the acceptance rate by fitting a geometric distribution as in Mamou, Jonathan, et al. "Accelerating Speculative Decoding using Dynamic Speculation Length." arXiv preprint arXiv:2405.04304 (2024).
>
> ---
>
> > Question 2: Why are you using an estimate of the acceptance rate instead of independently simulating acceptance/rejection for each drafter-target pair? For example, you could generate 'lookahead' tokens, verify them, go back to the drafter, and so on until the entire sequence is generated, and then use the estimates of TTFT and TPOT to estimate the total latency.
>
> The reported total latency and the total latency that is computed as suggested are equal in expectation.
>
> ---
>
> > Question 3: Please report the value of lookahead used for each row of Table 2, and clarify why that value was chosen.
>
> The lookahead is the minimal value out of 1, 5, and 10 that satisfies Equation 1, as mentioned in Lines 365-369.
>
> ---
>
> > Question 4: Why are real-world latencies of multithreading and context switching ignored (Page 10, lines 487-488)? These contribute to the overhead of DSI, and I feel that they should be reported and studied for a fair comparison with SI and non-SI.
>
> Table 2 reports speedups in a simulation where DSI is implemented as a multithreaded system that incurs all the real-world latencies, including context switching. (You referred to Figure 2, a visualization of an additional ablation experiment. This experiment implements an “offline” simulation where we count forwards and multiply them by their estimated latency.)
>
> ---
>
> > Question 5: Will it be possible to show any simulation results with multiple drafters? I feel that one of the strengths of DSI is that it can work with multiple drafters, and I would like to see how using multiple drafters compares with using a single drafter.
>
> Thank you. Adding drafters is mathematically guaranteed to increase the acceptance rate, as studied in SpecInfer. Since we use SpecInfer’s verification method as-is, their results immediately apply—suggesting the expected increase in the acceptance rate. Then, Figure 2 provides the expected speedup of DSI for any acceptance rate.
>
> *Miao, Xupeng, et al. "SpecInfer: Accelerating Generative Large Language Model Serving with Tree-based Speculative Inference and Verification." arXiv preprint arXiv:2305.09781 (2023).*
>
> ---
>
> We hope that our responses address your concerns and provide the necessary clarifications. We are grateful for your insightful comments, which have helped us improve our manuscript. Thank you once again for your time and consideration.

---

> > ### Comment · Reviewer_U6hk · 2024-11-28
> > **Re**
> >
> > Thank you for your response. I have a couple of follow-up questions:
> >
> > 1. Can you point me to the section of the paper Mamou, Jonathan, et al. "Accelerating Speculative Decoding using Dynamic Speculation Length." that you referred to for estimating the acceptance rate?
> >
> > 2. Can you justify/prove the claim made in response to Question 2 that "The reported total latency and the total latency that is computed as suggested are equal in expectation"?

---

> ### Author Response · Authors · 2024-11-28
>
> Thank you for following up and raising this important and thoughtful question.
>
> The *total latency* in this experiment is defined by the sums of the number of target and drafter forward passes. To calculate these sums, we simulate the process by sampling from the models, executing the verification procedure, and counting the number of required forwards. Since sampling from models is inherently stochastic, the total latency $T$ is a random variable, and we focus on its expectation, $\mathbb{E}[T]$.
>
> There exists an acceptance rate $p$ such that the expected latency satisfies $\mathbb{E}[T] = T$. This acceptance rate can be estimated in two ways. The first method directly computes the likelihood that a token is accepted by the verifier. Alternatively, as explained in Lines 407–410 of the paper, $p$ can be estimated by fitting a geometric distribution to the acceptance process. Specifically, this involves calculating the average number of tokens accepted per iteration and extracting $p$ as the parameter of the geometric distribution.
>
> Both approaches are consistent. As the number of iterations approaches infinity, the estimated acceptance rate converges to the true empirical acceptance rate $p$. Consequently, the estimated total latency based on $p$ converges to the actual total latency. This guarantees that the reported latency and the computed latency are equal in expectation.
>
> We greatly appreciate this question and recognize that additional clarity in this explanation would strengthen the paper. We will revise the manuscript to provide a more detailed and precise discussion of this point. Thank you again for your insightful feedback, which has helped us improve the presentation of our work.

---

> > ### Comment · Reviewer_U6hk · 2024-11-30
> > **Re**
> >
> > Thank you. Can you also point me to the section of the paper Mamou, Jonathan, et al. "Accelerating Speculative Decoding using Dynamic Speculation Length." that you referred to for estimating the acceptance rate on Page 8, line 410?

---

> > > ### Author Response · Authors · 2024-11-30
> > >
> > > Thank you for raising this follow-up question.
> > >
> > > Prior works in the field have developed theories and methods based on the simplifying assumption that token acceptance is independently and identically distributed (i.i.d.); for example, see [1]. In our paper, estimating the acceptance rate from the fitted geometric distribution is based on the same assumption. The results in [2] reveal that such an assumption is actually realistic. Figure 4 in [2] shows that the number of accepted tokens per iteration follows a geometric distribution, as explained below.
> > >
> > > [1]: Leviathan, Yaniv, Matan Kalman, and Yossi Matias. "Fast inference from transformers via speculative decoding." *International Conference on Machine Learning*. PMLR, 2023.
> > >
> > > [2]: Mamou, Jonathan, et al. "Accelerating Speculative Decoding using Dynamic Speculation Length." arXiv preprint arXiv:2405.04304 (2024).
> > >
> > > ---
> > >
> > > In speculative decoding, the number of new tokens generated by the $i$th forward pass of the target model (denoted by $n_i$) equals the Oracle Speculation Length (denoted by $\gamma_i^*$) plus one:
> > >
> > > $$
> > > n_i = \gamma_i^* + 1
> > > $$
> > >
> > > Figure 4 in [2] suggests that we can fit a geometric distribution to $n_i$ as follows:
> > >
> > > $$
> > > n_i = \min(\gamma_i + 1, \, g_i)
> > > $$
> > >
> > > where:
> > >
> > > - $\gamma_i$: The Speculation Length (SL) of the $i$th iteration.
> > > - $g_i \sim \text{Geometric}(\alpha)$, with $\alpha = \frac{1}{\text{average of } n_i}$.
> > >
> > > When modeling $n_i$ using $g_i \sim \text{Geometric}(\alpha)$, the expected acceptance rate is $\alpha$. We provide the fitted geometric distribution corresponding to the expected acceptance rates for Alpaca, CNN-DM, HumanEval, and MBPP. Since figures cannot be attached directly, we have shared the fitted geometric distributions via the link below:
> > >
> > > https://imgur.com/a/BNCyplI
> > >
> > > - **Alpaca:** Expected acceptance rate: 57.55%.
> > > - **CNN-DM:** Expected acceptance rate: 68.03%.
> > > - **HumanEval:** Expected acceptance rate: 90.47%.
> > > - **MBPP:** Expected acceptance rate: 91.68%.
> > >
> > > These acceptance rates support the validity of using a geometric distribution to estimate the acceptance rate, as they align with the empirical observations in [2].
> > >
> > > We hope this clarifies how we estimate the acceptance rate based on the method presented in [2], particularly in relation to Figure 4, which demonstrates that the actual number of accepted tokens per iteration distributes geometrically.
> > >
> > > Thank you again for your insightful question.

---

> > > > ### Comment · Reviewer_U6hk · 2024-12-03
> > > > **Re**
> > > >
> > > > Thank you for addressing all my concerns. I have increased my score.

---

### Author Response · Authors · 2024-11-23

**General Response to All Reviewers**

We sincerely thank all the reviewers for their insightful and constructive feedback. We are encouraged by the positive recognition of our work's novelty, theoretical rigor, and potential impact on accelerating LLM inference.

**Key Strengths Highlighted:**

- **Novelty and Contribution:**
  - *Reviewer U6hk:* "DSI appears to be a novel approach and effectively leverages the strengths of SI but also seeks to address its weaknesses..."
  - *Reviewer kPPx:* Our "proposed Speculation Parallelism effectively solves the blocking operation of speculative inference into non-blocking."
  - *Reviewer LmXh:* "The proposed speculation parallelism is novel and interesting, and might inspire future work on speculative inference or more general LLM inference acceleration."
  - *Reviewer 4xNe:* "The idea is clear and seems to work well through evaluation."

- **Theoretical Rigor:**
  - *Reviewer kPPx:* Noted that our "method is backed by proof."
  - *Reviewer LmXh:* Appreciated "the rigor in both theoretical analysis and numerical validation (via careful design of simulations)."

- **Empirical Validation:**
  - *Reviewer U6hk:* Observed that "the paper provides empirical evidence demonstrating that DSI outperforms both traditional speculative inference (SI) and non-SI methods."
  - *Reviewer 4xNe:* Mentioned that "experiments show speedups compared to the original speculative decoding."

- **Clarity and Presentation:**
  - *Reviewer kPPx:* Stated that the "paper is well written up to the method section."
  - *Reviewer 4xNe:* Found the paper "well-written and easy to follow."
  - *Reviewer LmXh:* Commented that "the paper is well-written and easy to follow."

- **Potential for Future Research:**
  - *Reviewer LmXh:* Mentioned that our work "might inspire future work on speculative inference or more general LLM inference acceleration."

**Revisions Made:**

- **Enhanced Evaluation Section:** We have improved the clarity of the evaluation section, providing more detailed explanations of our methodology and results to address concerns about the evaluations' strength and clarity, as suggested by *Reviewers kPPx* and *LmXh*.

- **Clarified Implementation Details:** In response to *Reviewers U6hk* and *LmXh*, we have provided additional explanations regarding the practical implementation of DSI, including KV cache management.

- **Addressed Resource and Deployment Concerns:** We have discussed the resource requirements and potential deployment challenges highlighted by *Reviewers U6hk* and *4xNe*, including the use of multiple GPUs and (energy) efficiency considerations.

- **Corrected Minor Issues:** We have fixed all identified typos and improved the writing for better flow and accuracy, as pointed out by *Reviewer LmXh*.

We believe these revisions address the concerns raised and enhance the overall quality of our work. We appreciate the reviewers' insights, which have significantly contributed to improving our paper.

Thank you once again for your time and consideration.

---

### Public Comment · ~Nadav_Timor1 · 2025-03-02

We sincerely thank the reviewers and the area chair for their thoughtful feedback and support. We’ve made updates to the camera-ready version to better address the raised concerns, including expanded explanations of the acceptance rate estimation method and KV cache management implementation.

---

### Meta-Review · Area_Chair_XQH2 · 2024-12-20

**Metareview:**

This work introduces a new method for parallel inference in large language models that improves speed, theoretical rigor, and clarity. Most reviewers agreed on its novelty, strong analysis, and clear writing, though some raised concerns about implementation complexity and energy use. After the authors explained their approach and provided more details, opinions shifted towards a more positive stance. While challenges remain in scaling and real-world deployment, the method’s potential speedups are promising. Overall, the paper’s novelty, in particular the theoretical analysis, and the overall contributions and responses justify acceptance. Minor improvments and further practical research are encouraged.

**Additional Comments On Reviewer Discussion:**

Reviewers initially raised concerns about: (1) acceptance rate estimation methodology, (2) evaluation strength and practical implementation feasibility, (3) KV cache management across GPUs, and (4) energy efficiency considerations. Authors provided detailed responses clarifying their acceptance rate calculations, justifying their simulation approach, and explaining how KV cache management would leverage existing solutions from SpecInfer. One reviewer increased their score after receiving clarification about the acceptance rate methodology. Another maintained their rating despite the authors' responses, while two others didn't comment further on score changes. Overall discussion focused heavily on technical implementation details and theoretical foundations. I weighted the reviews with lengthier engagements and discussion higher than the one-sided reviews in this borderline case.

---

### Decision · Program_Chairs · 2025-01-22

Accept (Poster)